# Enhancing CAR-T cell functionality in a patient-specific manner

David K. Y. Zhang [1,2], Kwasi Adu-Berchie [1,2,8], Siddharth Iyer[1,2,8], Yutong Liu[1,2], Nicoletta Cieri [3], Joshua M. Brockman[1,2], Donna Neuberg [4], Catherine J. Wu [3,5,6,7] & David J. Mooney [1,2] ✉

Patient responses to autologous CD19 chimeric antigen receptor (CAR) T-cell therapies are limited by insufficient and inconsistent cellular functionality. Here, we show that controlling the precise level of stimulation during T-cell activation to accommodate individual differences in the donor cells will dictate the functional attributes of CAR-T cell products. The functionality of CAR-T cell products, consisting of a diverse set of blood samples derived from healthy donors, acute lymphoblastic leukemia (ALL), and chronic lymphocytic lymphoma (CLL) patient samples, representing a range of patient health status, is tested upon culturing on artificial antigen-presenting cell scaffolds to deliver T-cell stimulatory ligands (anti-CD3/anti-CD28) at highly defined densities. A clear relationship is observed between the dose of stimulation, the phenotype of the T-cell blood sample prior to T-cell activation, and the functionality of the resulting CAR-T cell products. We present a model, based on this dataset, that predicts the precise stimulation needed to manufacture a desired CAR-T cell product, given the input T-cell attributes in the initial blood sample. These findings demonstrate a simple approach to enhance CAR-T functionality by personalizing the level of stimulation during T-cell activation to enable flexible manufacturing of more consistent and potent CAR-T cells.

Significant variability in the potency and durability of the antitumor response of autologous anti-CD19 chimeric antigen receptor (CAR)-T cells limits therapy success[1]. Previous analyses have identified critical determinants of response, including T-cell-extrinsic factors, such as the lymphodepletion regimen[2,3], the blood concentration of lactate dehydrogenase (LDH) or monocyte chemoattractant protein-1 (MCP-1)[2,3], and a host of T-cell-intrinsic determinants[3] aside from CAR construct design[1], including the CD4-to-CD8 ratio[4,5], and distinct T-cell subpopulations in the apheresis material[6] and in the CAR-T cell infusion product[7,8]. Knowledge of how to tune these functional attributes of the CAR-T product to improve therapeutic potency and durability

remains limited. The relationship between an apheresis (or blood) sample, the CAR-T manufacturing process, and the resulting cell product has not been extensively studied, let alone at a patient-specific level[9,10].

A critical step in CAR-T cell production is T-cell activation, which impacts the CAR-T transduction efficiency[11,12], rate of CAR-T cell expansion[13,14], and differentiation[15,16]. Synthetic artificial antigen presenting cells (aAPC) are often used to standardize the delivery of signal 1, T-cell receptor (TCR) stimulation (i.e., anti-CD3); and signal 2, costimulation (i.e., anti-CD28); while signal 3 (i.e., interleukin-2, IL-2), is generally supplemented exogenously. Dynabeads are widely used to

[1]John A. Paulson School of Engineering and Applied Sciences, Harvard University, Cambridge, MA, USA. [2]The Wyss Institute for Biologically Inspired Engineering, Harvard University, Cambridge, MA, USA. [3]Department of Medical Oncology, Dana-Farber Cancer Institute, Boston, MA, USA. [4]Department of Data Science, Dana-Farber Cancer Institute, Boston, MA, USA. [5]Department of Medicine, Brigham and Women's Hospital, Boston, MA, USA. [6]Harvard Medical School, Boston, MA, USA. [7]Broad Institute of MIT and Harvard, Cambridge, MA, USA. [8]These author contributed equally: Kwasi Adu-Berchie, Siddharth Iyer. ✉e-mail: mooneyd@seas.harvard.edu

manufacture CAR-T cell products, but they present anti-CD3 and anti-CD28 in a fixed manner (rather than from physiological fluid-like cell membranes), and at supraphysiological densities that can induce T-cell overstimulation and exhaustion[17–19]. Precisely tuning the stimulation dose (i.e., pg/cell) of signals 1 and 2 is not possible, as Dynabeads are administered at a specific bead:cell ratio. To overcome these limitations, we previously developed biodegradable, APC-mimetic scaffolds (APC-ms) to provide T-cell stimulation at highly-precise doses and in a physiological manner[20]. APC-ms presents T-cell activating ligands via a fluid lipid bilayer and slowly releases mitogenic factors to interacting T cells, mimicking critical features of natural antigen presentation (i.e., physical TCR rearrangement and paracrine signaling, respectively). To this end, APC-ms improves the quality (i.e., physiological context) of T-cell stimulation and enables more precise control of its strength (i.e., stimulation dose)[14,20].

Here, we investigate how CAR-T cell functionality can be enhanced by fine-tuning the dose of stimulation during T-cell activation. APC-ms are used to precisely control the density of anti-CD3/anti-CD28 presented to a diverse set of T-cell samples representing varying patient health status, including healthy donor T-cells, and T-cells of acute lymphoblastic leukemia (ALL) and chronic lymphocytic lymphoma (CLL) origin, to produce a diverse library of CAR-T cell products. The products are phenotypically and functionally distinct across stimulation dose and health status. By modeling the relationship between blood T-cell samples and CAR-T cells, we identify T-cell stimulation dose to be a critical parameter in modulating CAR-T cell functionality. We further demonstrate a proof-of-concept prediction model that outputs an APC-ms stimulation dose needed to obtain a specific CAR-T cell product in a patient-specific manner, based on the input T-cell blood sample phenotype. Personalizing stimulation dose during T-cell activation represents a simple strategy that enables more consistent manufacturing of CAR-T cells while potentiating their therapeutic activity.

## Results
### Characterization of T cells isolated from healthy and patient samples and their CAR-T cell products
Peripheral blood mononuclear cells (PBMC) were obtained from blood samples from eight healthy donors, six patients with high-risk acute lymphoblastic leukemia (ALL), and four patients with chronic lymphocytic lymphoma (CLL) (Supplementary Table 1). PBMCs from healthy donor and patient blood samples were phenotyped via flow cytometry using a standard FACS panel and gating strategy (Supplementary Fig. 1a). Principal component analysis (PCA) revealed a patient health status-dependent separation in T-cell phenotype (Fig. 1a and Supplementary Fig. 2a). No notable differences between ALL vs CLL samples were observed; therefore, ALL/CLL were grouped for analyses and referred to as "patient/patient-derived" samples. Although similar frequencies in CD3+ T cells were observed (Fig. 1b), T cells from patient-derived samples were significantly CD8-biased (Fig. 1c) and contained significantly more CD45RA+CCR7−, suggesting a more effector-biased phenotype (Fig. 1d). No significant differences were observed in the expression of PD-1, TIM-3, CD25, and CD137 between blood samples (Supplementary Fig. 2b–e), aside from a minor trend with PD-1 expression in CD3+ T cells (Supplementary Fig. 2c). Patient-derived CD4+ T cells expressed more CD95+ than naive healthy donor CD4+ T cells, suggesting a more differentiated, stem memory-like phenotype than naive healthy donor CD4+ T cells (Supplementary Fig. 2f).

We produced anti-CD19 CAR-T cells using cells from all samples using a second-generation construct consisting of an extracellular anti-CD19 scFv linked to an intracellular 41BBζ signaling domain. Isolated CD3+ T cells were activated with either Dynabeads or APC-ms across a range of stimulation doses (Fig. 1e and Supplementary Table 2). Under these cell and material co-culture conditions, APC-ms presenting anti-CD3/anti-CD28 at a density corresponding to 0.1 mol% biotin

(abbreviated A0.1) is stimulation dose-matched to Dynabeads seeded at a 3:1 bead:cell ratio (abbreviated D3:1, Fig. 1f), a commonly-recommended Dynabead dosing[20]. While IL-2 can be sustainably released from APC-ms and leads to greater T cell expansion than the medium supplementation required with Dynabeads[20] (Supplementary Fig. 3a), IL-2 was supplemented in the medium for both conditions to facilitate a fully matched comparison of signals 1–3.

As expected, lower expansion (Fig. 1g) was observed in CAR-T cells derived from patient samples compared to healthy donors at the highest proliferation-inducing stimulation conditions ($p = 0.05$). When directly comparing APC-ms and Dynabead dose-matched conditions, we observed that A0.1 appeared to promote more expansion than D3:1 in patient-derived but not healthy donor samples. A trend suggesting lower fold expansion was observed with increasing stimulation dose in patient-derived, but not healthy samples. In healthy donor CAR-T cells, we did not observe any significant differences in transduction efficiency (Supplementary Fig. 3b). In patient-derived CAR-T cells, we observed a stimulation dose-dependent effect CAR-expression−A0.3 CAR-T cell products expressed more CAR than A0.1 CAR-T cell products (Supplementary Fig. 3b). Notably, increasing the stimulation dose from A0.1 to A0.3 corresponded with progressively higher levels of PD-1 and TIM-3 co-expression among both CD4+ and CD8+ T cells (Fig. 1h), and higher CD4:CD8 ratios (Supplementary Fig. 3c) and stimulation dose-dependent changes in CD45RA and CCR7 expression (Fig. 1i). We observed a reduction in patient-derived CD45RA+CCR7+ CD8+ T cells with increasing stimulation beyond the A0.15 stimulation dose and what appeared to be a corresponding reduction in CD45RA+CCR7− CD8+ T cells, suggesting fewer effector-like and stem memory-like T cells[21] (Supplementary Fig. 3d and Fig. 1i). Increasing stimulation dose also led to increased expression of CD25 and CD137 in both CD4+ and CD8+ T cells (Supplementary Fig. 3e, f). Comparing dose-matched groups, Dynabeads (D3:1) tended towards a more activated and exhausted phenotype compared to APC-ms (A0.1). We observed stimulation dose as a driving factor in generating large phenotypic changes in CAR-T cell products by evaluating their global differences (Fig. 1j). These observations highlight the differences between healthy donor and patient-derived samples prior to and after CAR-T cell production. Moreover, they demonstrate the importance of stimulation dose in significantly skewing T-cell phenotype towards a highly-activated (i.e., CD25hi, CD137hi), less effector-like (CD45RA−) phenotype, and the differences between Dynabeads and APC-ms stimulation under signals 1-3-matched conditions.

### CAR-T cells engineered from healthy samples are generally more functional than those derived from patient samples
Patient-derived CAR-T cells expressed less IL-2 than healthy CAR-T cells following Raji-cell stimulation, regardless of stimulation type (Dynabeads vs APC-ms) or dose (Fig. 2a, b). The expression of IL-2 following tumor antigen exposure in CD8+ T cells has been associated with longer-surviving and more potent CAR-T cells[22]. Increasing stimulation dose did not result in any significant differences in IL-2 production in CD8+ T cells (Fig. 2b and Supplementary Fig. 4a). Moreover, no significant differences were observed in the expression of TNF-α, IFN-γ, and granzyme-β within and between healthy donor CAR-T cells and patient-derived CAR-T cells (Supplementary Fig. 4b–d). A trend suggesting that increasing stimulation dose promoted granzyme-β expression in patient-derived CD8+ CAR-T cells but not healthy CD8+ CAR-T cells was observed (Supplementary Fig. 4d). Interestingly, when comparing Dynabeads and APC-ms groups, healthy donor-derived A0.1 CAR-T cell products tended to express more IL-2 and TNF-α than their D3:1 CAR-T cell counterparts (Fig. 2b and Supplementary Fig. 4b). Additionally, similar to the trends observed in CD8+ CAR-T cells, healthy CD4+ CAR-T cells expressed more IL-2 than patient-derived CD4+ CAR-T cells (Supplementary Fig. 4e).

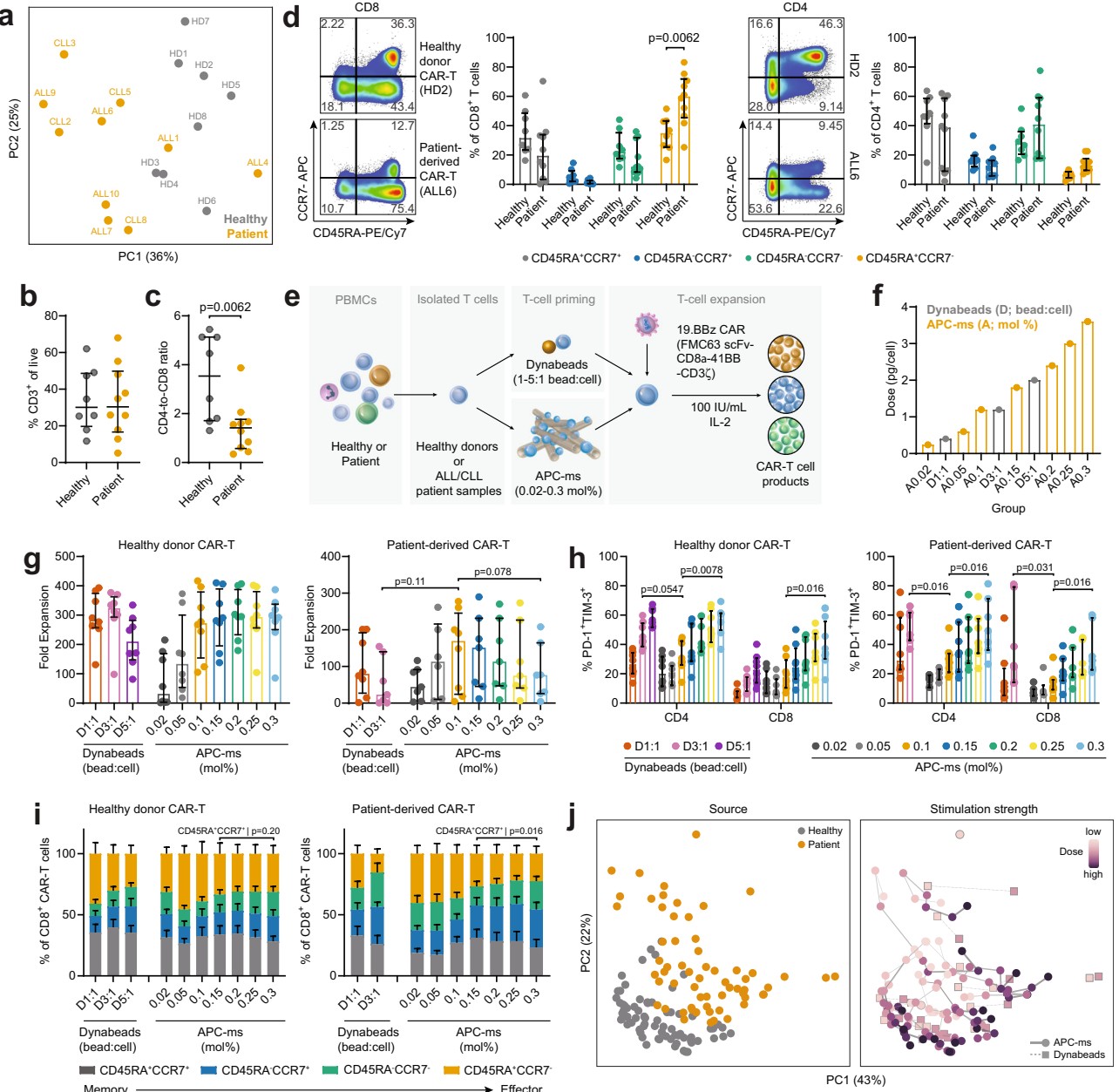

**Fig. 1 | Stimulation strength and quality during T-cell activation drives the production of distinct CAR-T cell products. a** Principal component analysis (PCA) of blood samples stained using a standard 12-channel T-cell phenotyping FACS panel (Supplementary Fig. 1a) and colored by sample source in the PCA plot (healthy donor, HD; acute lymphoblastic leukemia, ALL; chronic lymphocytic leukemia, CLL). Frequency of CD3$^+$ T cells (**b**), and CD4-to-CD8 ratio (**c**), in healthy or patient-derived PBMC samples. **d** Proportions of CD8$^+$ and CD4$^+$ T cells stained for T-cell memory markers CD45RA and CCR7, highlighting increased fractions of CD45RA$^+$CCR7$^-$ CD8$^+$ T cells in patient PBMC samples. Representative FACS plots for a healthy donor and patient sample are shown. **e** Schematic overview of CAR-T cell production pipeline. Isolated T cells were activated with artificial antigen presenting cell-mimetic scaffold (APC-ms) across a progressively increasing range of stimulation dose (0.02–0.3 mol%) or Dynabeads at a 1:1, 3:1, or 5:1 bead:cell ratio. Cells were transduced to express a second-generation CD19 CAR construct and expanded using exogenous IL-2 supplementation. **f** Anti-CD3/anti-CD28 stimulation measured in pg/cell across various APC-ms and Dynabead

groups. Fold expansion (**g**) and PD-1, TIM-3 co-expression (**h**) of day 8-expanded CAR-T cell products. The Dynabead 5:1 condition was not used to stimulate the patient-derived T-cell samples. **i** CD45RA and CCR7 expression among healthy and patient-derived CD8$^+$ CAR-T cells. (**j**) PCA of CAR-T cell products produced via APC-ms and Dynabeads, colored by source/T-cell state (healthy vs patient-derived; left), and stimulation strength (right). CAR-T cells derived from patient samples are phenotypically distinct from healthy donor samples. Data represent median ± interquartile range of different healthy and patient-derived samples. Data in **b**–**d** represent n = 8 healthy donors and n = 10 patient samples. Data in **g**–**i** represent n = 8 healthy donor-derived CAR-T cell products and n = 8 patient sample-derived CAR-T cell products. One patient sample failed to expand regardless of stimulation type. For a separate patient sample, only A0.1 and D1:1 groups were evaluated due to limited starting material. Comparisons in **c**, **d** were calculated using two-sided Mann–Whitney tests, while comparisons in **g**–**i** were calculated using two-sided Wilcoxon signed-rank tests. Comparisons between healthy donor and patient-derived cells calculated using two-sided Mann–Whitney tests.

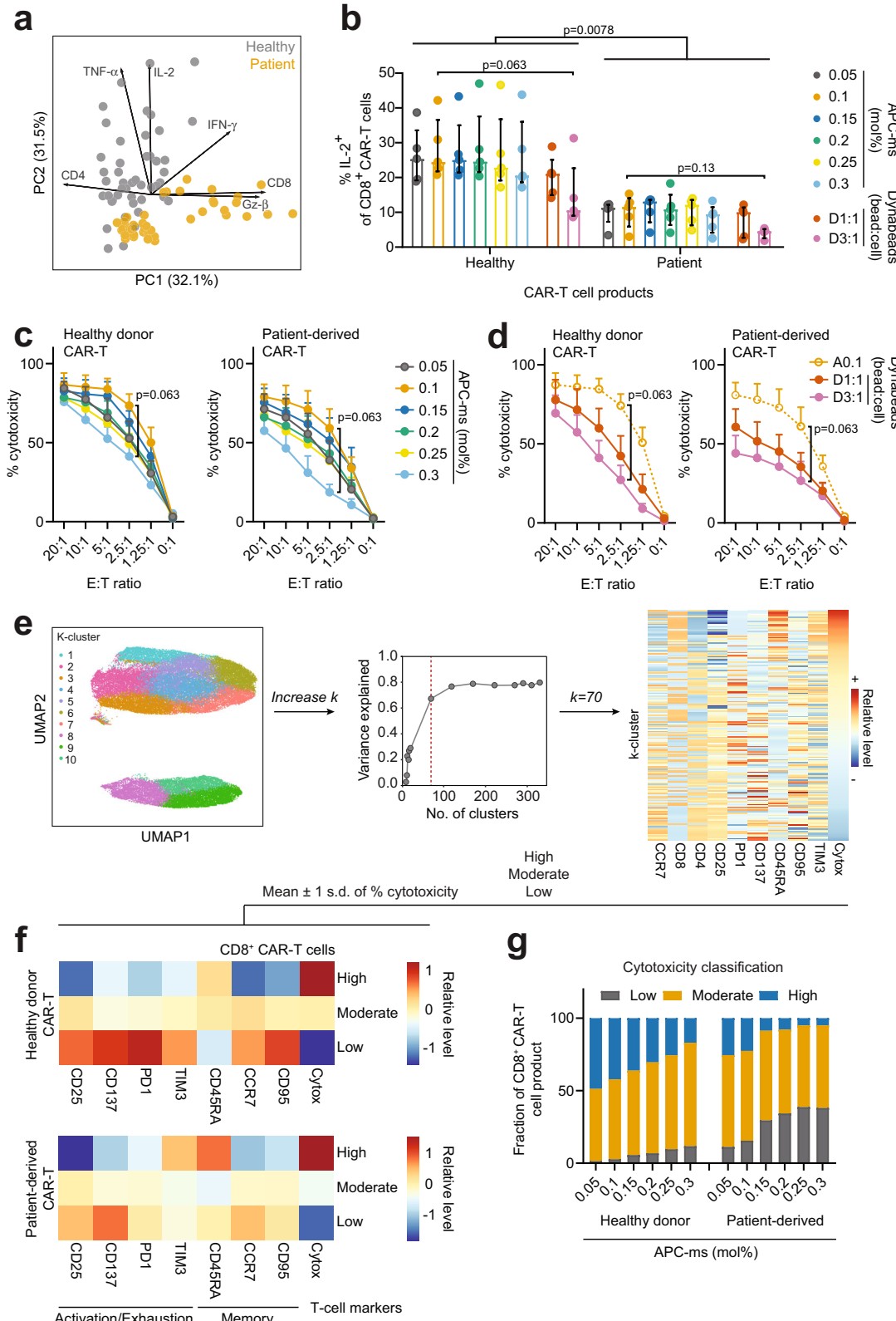

The ex vivo antitumor potencies of healthy donor and patient-derived CAR-T cells were next compared. CAR-T cells derived from patient samples elicited overall lower killing regardless of the type of stimulation (e.g., APC-ms or Dynabeads) (Fig. 2c, d and Supplementary Fig. 4f); however, stimulation dose impacted the cytotoxic potential of the CAR-T cells. For Dynabeads, activation with D3:1 resulted in a trend suggesting less cytotoxic CAR-T cells than D1:1 (Fig. 2d). For APC-ms,

despite having more CD45RA⁺CCR7⁻ CD8⁺ CAR-T cells in the A0.05 vs A0.1 product, relatively more Raji-cell killing was observed in A0.1 products regardless of patient health status, suggesting that the A0.05 products may be under-stimulated (Fig. 2c and Supplementary Fig. 4f). Increasing APC-ms stimulation dose from 0.1 to 0.3 mol% appeared to produce progressively less cytotoxic CAR-T cell products, in both healthy donor and patient-derived samples.

**Fig. 2 | CAR-T cells derived from healthy and patient samples show differential effector functions following in vitro antigen exposure in a stimulation-dependent manner. a** PCA of cytokine expression in CAR-T cell products following 1.25:1 (effector: target) stimulation with Raji cells, colored by disease state. Arrows represent PCA loadings and magnitude of contribution to PC1 and PC2.
**b** Frequency of IL-2-expressing CD8$^+$ CAR-T cells generated from healthy and patient samples across varying stimulation doses following 1.25:1 (effector: target) stimulation with Raji cells. In vitro cytotoxicity of CAR-T cell products generated across a range of APC-ms (**c**) or Dynabead (**d**) stimulation following co-culture with Raji-luc cells. Left: healthy donor CAR-T, right: patient-derived CAR-T. **e** Schematic representation of workflow linking CAR-T cell cytotoxicity and phenotype. Single cell fluorescence data from healthy donors or patient-derived samples (a patient sample is shown) were aggregated and clustered using $K$-means clustering with progressively increasing $k$ to maximize the clustering explained variance, then

visualized in a heatmap. Cytotoxicity was assigned to each single cell within each CAR-T cell product, normalized and thresholded at mean ± 1 s.d. to represent low, moderate, and high cytotoxicity, as well as the average expression of CAR-T cell phenotypic markers associated with these levels. The % cytotoxicity was taken at the 1.25:1 effector: target (E:T) ratio. **f** Heatmaps of CD8$^+$ CAR-T cell markers associated with either high, moderate, or low levels of cytotoxicity in healthy and patient-derived CAR-T cells. **g** Fraction of cells classified as high, moderate, or low cytotoxicity mapped to APC-ms stimulation dose. Data represent median ± interquartile range. Data in **b**–**d**, **g** represent CAR-T cell products derived from $n = 5$ healthy and $n = 5$ patient samples. For the D3:1 group, $n = 4$ patient samples were assessed. Comparisons in **b** were calculated using either two-sided Mann–Whitney or Wilcoxon signed-rank tests for comparisons between healthy vs patient CAR-T or within either healthy or patient CAR-T, respectively. Comparisons in **c**, **d** were calculated using two-sided Wilcoxon signed-rank tests.

To explore the relationship between stimulation, CAR-T cell phenotype, and cytotoxic potential at a global level, we devised a clustering-based strategy using the single-cell fluorescence data from our flow cytometry phenotyping analyses for APC-ms-activated CAR-T cell products. For each CAR$^+$CD3$^+$ cell event, we assigned a % cytotoxicity value (from studies using 1.25:1 E:T) for each healthy donor or patient-derived CAR-T cell product and performed K-means clustering at progressively increasing $k$ (Fig. 2e). Then, for each $k$, a random forest regression was performed to predict cluster-level cytotoxicity, with the goal of selecting the optimum number of clusters that best explains cluster-level cytotoxic potential. Three levels of cytotoxicity are defined to represent low, moderate, and high % cytotoxicity, thresholded at mean ± 1 s.d, as well as the average expression of CAR T-cell phenotypic markers associated with these levels, and visualized as a heatmap (Fig. 2f). Comparing healthy and patient cells, similar T-cell markers were predicted to represent highly-cytotoxic CAR-T cells, except for TIM-3 expression, which was elevated in patient-derived CAR-T cells (Fig. 2f, bottom). CD4$^+$ CAR-T cells were not predicted to be robust killers (Supplementary Fig. 4g). We chose Random Forest regression over linear regression or GAM since the latter two assume linearity. GAM models showed similar results (Supplementary Fig. 4h, i). A linear, progressive loss in healthy highly cytotoxic CAR-T cells were observed with increasing APC-ms stimulation dose (Fig. 2g). However, in patient-derived CAR-T cells, a non-linear relationship was observed with increasing APC-ms stimulation dose, suggesting that patient-derived T cells may have a lower activation threshold than healthy donor T cells.

## T-cell stimulation dose regulates in vivo CAR-T cell responses

We next investigated how cell-intrinsic differences in the CAR-T cell product manifest in vivo at the intrapatient level. Since APC-ms enabled greater fine-tuning of CAR-T cell phenotype and function than Dynabeads for patient-derived CAR-T cell products, we focused on APC-ms-generated products. Three CAR-T cell products were selected from an ALL patient; A0.1, A0.15, and A0.3, based on their substantial differences in CAR-T cell phenotype and cytotoxic potential (Fig. 3a, b). A progressive loss in cells classified as highly cytotoxic was noted with increasing stimulation (Fig. 3b). Moreover, the A0.15 product contained more CD4$^+$ CAR-T cells (Fig. 3c) and more PD-1$^+$TIM-3$^+$ CAR-T cells than A0.1 product (Fig. 3d). Increasing APC-ms stimulation dose led to a progressive loss in CD45RA$^+$CCR7$^-$ CD4$^+$ and a sharp decrease in the number of CD8$^+$ CAR-T cells (Fig. 3e). The CAR-transduction efficiency was similar between the A0.1 and A0.15 products but was increased in the A0.3 CAR-T cell products (Supplementary Fig. 5a).

The three CAR-T cell products were tested in a disseminated xenograft model of Burkitt's lymphoma using CD19-expressing Raji-luc cells (Fig. 3f). While there were substantially fewer CD45RA$^+$CCR7$^-$ CD8$^+$ effector-like CAR-T cells in the A0.15 product relative to A0.1 product, both led to similar degrees of tumor control (Fig. 3g, h). In contrast, most animals treated with the A0.3 product succumbed to

the tumor, accompanied by dramatic losses in weight (Supplementary Fig. 5b). Analysis of the evolution of the CAR-T cell products revealed faster rates of in vivo CAR-T cell expansion in the A0.15 and A0.3 products compared to the A0.1 product following infusion (Fig. 3i). Whereas in animals treated with the A0.1 product, the circulating number of CAR-T cells increased more slowly and steadily (Fig. 3i). In animals treated with the A0.15 product, the concentration of CAR-T cells peaked at day 9 and progressively declined, while in animals treated with the A0.3 product, the concentration of CAR-T cells remained elevated (Fig. 3i). Some of these differences could be explained by massive non-specific expansion of CD3$^+$ T-cells, particularly in animals treated with the A0.1 product (Fig. 3j and Supplementary Fig. 5c). Furthermore, several of these animals (4/7 in the A0.1 group, 1/7 in the A0.15 group, and 0/7 in the A0.15 group) experienced significant hair loss consistent with symptoms of xenogeneic graft-vs-host disease (GVHD; Supplementary Fig. 5d).

Since there were low numbers of T cells collected at some time and since no significant differences were noted in the expression of any marker between transferred CAR$^+$ and CAR$^-$ cells, we did not distinguish between CAR$^+$ and CAR$^-$ subpopulations for subsequent phenotypic analysis (Supplementary Fig. 5e). Distinct patterns in the evolution of CD45RA and CCR7 T-cell subpopulations were observed (Fig. 3k and Supplementary Fig. 5f). While T cells in all conditions became increasingly more CD45RA$^+$CCR7$^-$ effector-like over time, few CD45RA$^+$CCR7$^+$ T cells were found in animals treated with the A0.3 product, suggesting few stem-memory CD8$^+$ T cells. In A0.1 product-treated animals, a large proportion of CD45RA$^+$CCR7$^+$ T cells was found but declined longitudinally giving way to CD45RA$^+$CCR7$^-$ T cells. In A0.15 product-treated animals, we observed fewer CD45RA$^+$CCR7$^+$ T cells and a slower enrichment of CD45RA$^+$CCR7$^-$ T cells relative to animals treated with the A0.1 product. Higher proportions of PD-1$^+$TIM-3$^+$ CD8$^+$ T cells were found in A0.3 product-treated animals relative to the A0.1 and A0.15 products (Fig. 3l). Similar trends were observed in CD4$^+$ T cells (Supplementary Fig. 5g). A0.3 product-treated animals also tended to have higher proportions of CAR-expressing T cells, which could be attributed to continuous antigen exposure as a result of tumor cell persistence (Supplementary Fig. 5h). Consistent with the pre-infusion product phenotype, the animals treated with the A0.3-product had the highest CD4:CD8 ratio, which declined rapidly over time (Fig. 3m). In A0.1 product-treated animals, a CD4-bias peaking was observed ~17 days post CAR-T cell infusion, suggesting in vivo CD4$^+$ CAR-T cell expansion. A similar trend was observed in animals treated with the A0.15 product, albeit to a greater magnitude, peaking ~21 days post infusion (Fig. 3m).

Similar trends in CAR-T cell responsiveness were observed using A0.1 and A0.3 products generated from a separate healthy donor T-cell sample (Supplementary Fig. 6). Increasing the administered CAR-T cell dosage of the A0.3 product 3-fold, from $5 \times 10^5$ to $1.5 \times 10^6$ did not recover the therapeutic outcomes of the A0.1 product (Supplementary Fig. 6a–g). When comparing the A0.1 product and its dose-matched

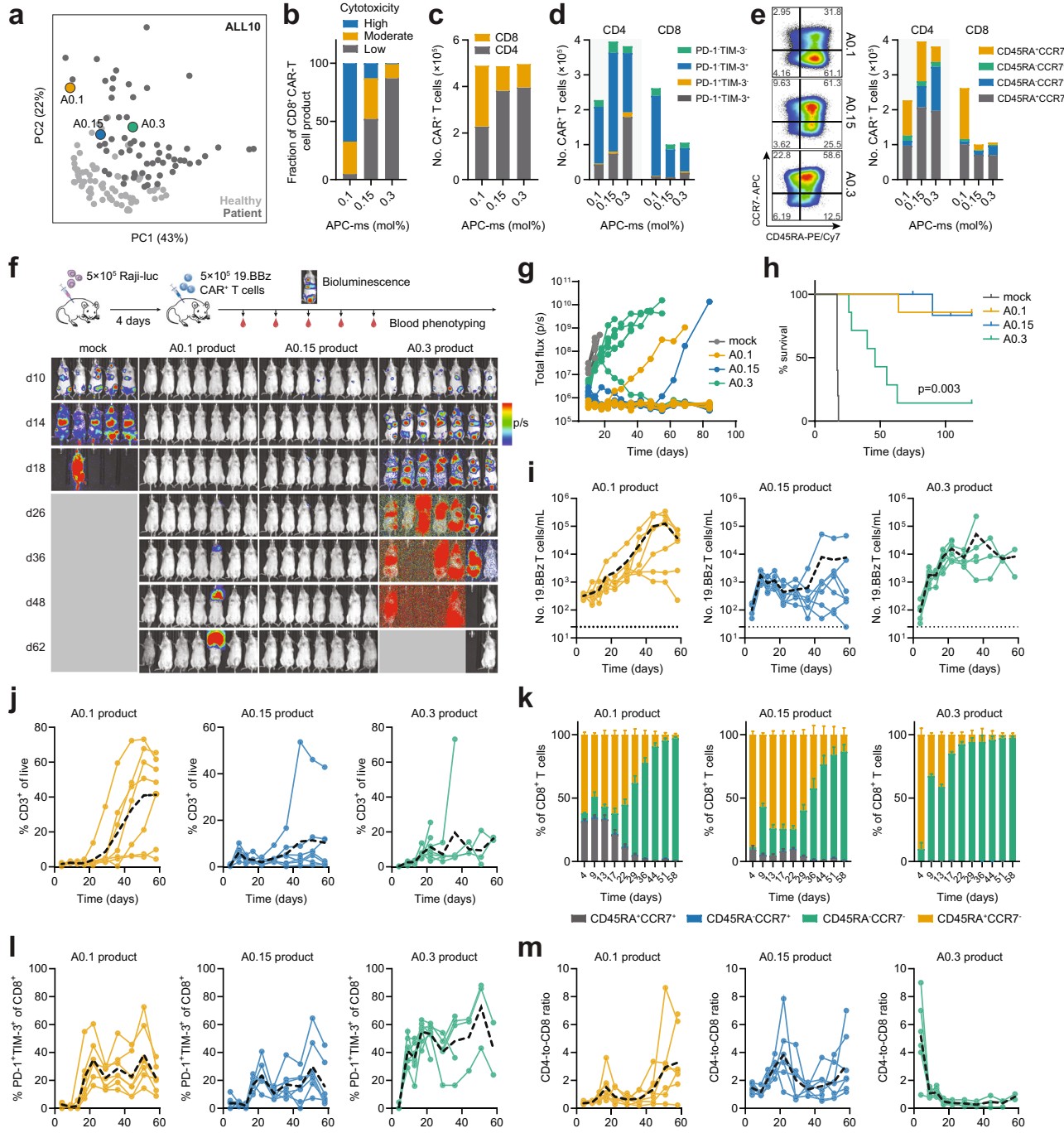

**Fig. 3 | Stimulation dose during T-cell activation tunes CAR-T cell responsiveness in a xenograft lymphoma model. a** Selection of APC-ms-primed CAR-T cell products for in vivo analysis. Three CAR-T cell products (0.1 mol%, 0.15 mol%, and 0.3 mol% APC-ms; abbreviated A0.1, A0.15, and A0.3, respectively) derived from an ALL sample (ALL10) were used. Properties of the selected CAR-T cell products, including cytotoxicity classification (**b**), the number of CD4+ or CD8+ cells (**c**), PD-1 or TIM-3 expressing T cells (**d**), and CD45RA or CCR7-expressing T cells (**e**). Representative FACS plots showing CD45RA and CCR7 expression among CD8+ T cells shown in **e**; left. **f** Study outline of disseminated Raji xenograft model and bioluminescent images showing tumor burden. $5 \times 10^5$ luciferized Raji cells were administered four days prior to treatment with $5 \times 10^5$ CAR+ T cells. CAR-T cell products generated via APC-ms from ALL10 were used for the animal study: mock,

$n = 5$; A0.1 (0.1 mol% APC-ms stimulated CAR-T cells), $n = 7$; A0.15 (0.15 mol% APC-ms stimulated CAR-T cells), $n = 7$; A0.3 (0.3 mol% APC-ms stimulated CAR-T cells), $n = 7$ mice. Quantification of bioluminescent signal (**g**) and animal survival (**h**). Mice were bled at the indicated time points, and blood cells were processed and analyzed by flow cytometry. **i** In vivo concentration of circulating CAR-T cells following infusion. **j** Change in frequency of CD3+ T cells following infusion. Change in CD45RA or CCR7 expression in CD8+ T cells (**k**), PD-1/Tim-3 co-expression in CD8+ T cells (**l**), and CD4-to-CD8 ratio (**m**) following CAR-T cell infusion. Dotted black curves in **i**, **j** and **l**, **m** represent averages. Data in **k** represents mean ± s.e.m. and one experimental replicate. Animal survival in **h** was calculated using a two-tailed log-rank Mantel−Cox test. Two animals in the A0.15 group were excluded due to non-tumor-related complications.

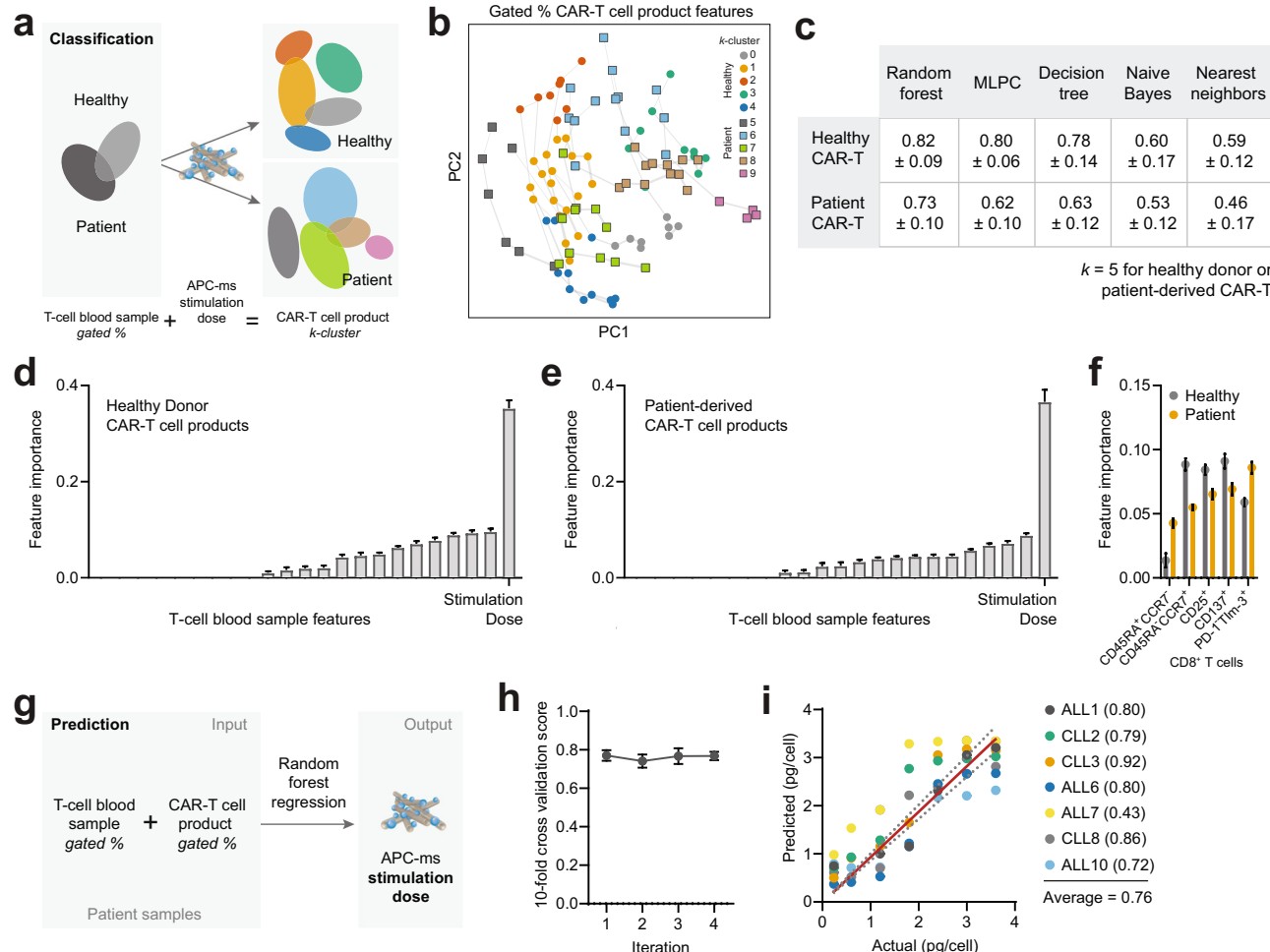

**Fig. 4 | A model mapping CAR-T cell products to T-cell blood sample features and stimulation dose. a** Schematic of classification model. The relationship between the phenotype of a T-cell blood sample, APC-ms stimulation dose, and its resulting CAR-T cell product(s) was quantified separately for healthy donor and patient-derived samples. **b** K-means clustering performed separately on healthy and patient-derived CAR-T cell products. The number of k-clusters was set as 5. A standard 12-marker FACS panel and gating strategy were used to establish features in the T-cell blood sample (see Supplementary Fig. 1b for gating strategy). **c** Cross-validated prediction scores of various machine learning classification algorithms. Different classifiers were iteratively trained on a subset of the data to predict the CAR-T cell product k-cluster based on the T-cell phenotype of the blood sample and stimulation level—the predictions were tested on the remaining subset of data.

Feature importance of various T-cell intrinsic features in the blood sample in healthy donors (**d**) and patient samples (**e**) using the random forest classifier. **f** Top five most important features in the model for healthy donor and patient-derived samples, excluding stimulation dose. **g** Schematic of prediction model. The model predicts an APC-ms stimulation given a desired CAR-T cell phenotype and input T-cell blood sample phenotype. **h** Tenfold cross-validation scores for patient-derived samples using random forest regression. **i** Predicted vs actual stimulation values (pg/cell) using random forest regression for patient-derived patient samples. For each patient sample, the prediction accuracy, represented by the $R^2$ between predicted and actual APC-ms stimulation doses following sample hold-out, is listed. The dotted gray lines represent the 95% confidence bands. Data in **d**–**f**, **h** represent mean ± s.d.

D3:1 product, we observed the A0.1 product showed improved in vitro cytotoxic function than their Dynabead counterparts (Supplementary Fig. 6h), which translated to improved animal survival in vivo (Supplementary Fig. 6i, j). These data demonstrate the capacity of fine-tuning T-cell activation strength to manipulate in vivo CAR-T cell behaviors and functionality.

## A model for predicting T-cell stimulation dose for personalized CAR-T cell production

To extend the utility of these findings, we quantified the relationship between a T-cell blood sample and its CAR-T cell product(s) using a classification model. Then, as a proof-of-concept, we built a prediction model that outputs a required APC-ms stimulation dose to achieve a desired CAR-T cell product phenotype at a patient specific level, given their input T-cell phenotype in the blood sample (Fig. 4a). We used gated T-cell frequencies (instead of MFIs in previous figures) defined in our conventional gating strategy as input T-cell blood sample features

(Supplementary Fig. 1b) to broaden the utility of the model to routine and conventional phenotyping gates.

Consistent with the observation that CAR-T cells derived from healthy and patient samples were phenotypically distinct (Fig. 1j, left panel), a random forest machine learning (ML) classifier identified with >90% accuracy whether a CAR-T cell product was derived from healthy or patient T cells (Supplementary Fig. 7a) based on phenotype alone. Therefore, separate classification models were built for healthy and patient-derived samples. K-means clustering was performed on healthy donor and patient-derived CAR-T cells separately to group phenotypically similar CAR-T cell products across stimulation dose and healthy donor/patient sample (Fig. 4b); k = 5 could consistently explain >90% of the variance, and was selected for the following analyses (Supplementary Fig. 7b). Some clusters contained CAR-T cell products derived from a single T-cell blood sample, while others contained products derived from multiple blood samples (Fig. 4b). Then, we asked if the model could be

used to classify the k-cluster a specific CAR-T cell product would belong to, given only the input T-cell phenotype and stimulation dose. At $k = 5$, we found high (>70%) 10-fold cross-validation scores in most ML classifiers tested, demonstrating that the relationship between the T-cell blood phenotype, APC-ms stimulation dose, and the CAR-T cell k-cluster phenotype is quantifiable and predictable (Fig. 4c).

Using the trained random forest ML classifier, which we chose to examine due to the model's interpretability and high scoring in both healthy donor and patient samples, we determined the relative importance of each T-cell feature in either healthy or patient-derived T-cell blood samples. APC-ms stimulation dose was determined to be the most important contributing feature to the classifier in both healthy and patient-derived samples (Fig. 4d, e). However, the phenotype of the input T cells, and whether it came from a healthy donor or patient sample is also important—models trained only on stimulation dose were significantly worse in accuracy (Supplementary Table 4). In patient-derived samples, the fraction of PD-1⁻TIM-3⁺ CD8⁺ T cells was the second most important feature following stimulation dose, followed by the fraction of CD25⁺ and CD137⁺ CD8⁺ T cells, and CD45RA⁻CCR7⁺ and CD45RA⁺CCR7⁻ CD8⁺ T cells. Whereas in healthy samples, the order of feature importance was different, suggesting that healthy and patient-derived T cells have distinct cell-intrinsic features that are important for CAR-T cell production (Fig. 4f). The top two most important features following APC-ms stimulation dose (i.e., PD-1⁻TIM-3⁺ and CD137⁺ CD8⁺ T cells) were unchanged at various k (Supplementary Fig. 7c).

We next explored the utility of the relationship between a blood sample and its CAR-T cell product in a proof-of-concept prediction model. The model predicts a specific APC-ms stimulation dose, given a desired CAR-T cell phenotype and input T-cell blood sample phenotype (Fig. 4g). Such a model would be useful for guiding CAR-T cell production when desired CAR-T cell-intrinsic features are known. We focused on building the prediction model specifically for patient samples, as these are more clinically-relevant, particularly in the autologous setting. Robust cross-validation scores were observed using random forest regression (Fig. 4h). To validate the model's predictive capabilities, we held out one set of patient-derived CAR-T cell products at a time and asked the model to predict the APC-ms stimulation dose required given the input T-cell blood sample phenotype and CAR-T cell phenotype at each APC-ms stimulation level for the held-out patient. We observed an average 0.76 correlation score for the patient-derived samples (Fig. 4i). Certain patient samples were predicted with very high accuracy (e.g., 91%), while the accuracy was considerably lower with ALL7 (43%), which was observed to be an outlier in the dataset (Supplementary Fig. 7b). Taken together, this data demonstrates that the relationship between the initial T-cell blood sample and the CAR-T cell product is predictable; highlighting the utility of tuning T-cell stimulation strength to personalize CAR-T cell manufacturing to improve CAR-T cell functionality and therapeutic outcomes.

## Discussion

Our findings demonstrate that stimulation dose during T-cell activation has profound effects on the resulting phenotype and function of CAR-T cell products. Using a synthetic aAPC that enables the precise delivery of anti-CD3/anti-CD28 stimulation via a physiological fluid bilayer, we generated a library of CAR-T cell products from healthy or patient (i.e., ALL or CLL) blood samples. Studying the link between a sample and its CAR-T cell product(s) revealed a quantifiable and predictable relationship that is dependent on the dose of stimulation during T-cell activation and also the disease status of the T-cell blood sample (i.e., healthy vs ALL/CLL), including the cumulative history of past medical treatments[23,24]. Furthermore, this relationship and our proof-of-concept prediction model highlights stimulation dose as a

tool for generating more consistent and highly functional autologous CAR-T cell products.

A striking difference was found in T-cell activation potential and antitumor proficiency of T cells collected from healthy donor and heavily pretreated hematological patient samples. Within the patient cohort however, we did not observe substantial phenotypic differences between T cells derived from ALL or CLL samples, despite being obtained from patients with different treatment histories and disease stages; we cannot rule out that there might be distinctions between ALL- and CLL-derived T cells that could be resolved by increasing the number of patient samples studied and/or through higher resolution characterizations.

Controlling the dose of polyclonal anti-CD3/anti-CD28 stimulation during T-cell activation profoundly changed the resulting CAR-T cell phenotype and function. Under dose (i.e., signals 1, 2, 3)-matched conditions, APC-ms (A0.1)-treated patient-derived T cells trended toward greater proliferation compared to Dynabeads (D3:1), despite not benefiting from pre-loaded IL-2[20], suggesting that delivering T-cell ligands under physiologically relevant contexts (i.e., via a lipid bilayer) was desirable, particularly for patient-derived T cells. Pre-loading IL-2 into APC-ms improves T-cell proliferation by providing paracrine signaling, mimicking the in vivo context. Interestingly, our observations indicated that patient-derived T cells may have a lower activation threshold than healthy donor T cells. Patient-derived T-cell blood samples contained significantly more CD8⁺ T cells that were CD45RA⁺CCR7⁻, suggesting a more effector-like phenotype compared to healthy blood samples[25], and effector T cells may have lower activation thresholds compared to their central memory and stem memory counterparts[26-28]. However, while we observed a linear, progressive decline in the fraction of highly cytotoxic T-cells with increasing stimulation dose in healthy samples, we observed a nonlinear decline in the fraction of highly cytotoxic cells in patient-derived samples that accelerated beyond 0.1 mol% APC-ms stimulation, supporting the notion that these T cells are more sensitive to stimulation. High levels of anti-CD3/anti-CD28 (e.g., D3:1, A0.2 and higher) may have caused overstimulation and activation-induced cell death (AICD) in patient-derived CAR-T cells owing to their enriched fractions of CD8⁺ effector cells. The functional and cytotoxicity classification analyses further support this possibility.

Unique in vivo behaviors were observed with three distinct CAR-T cell products derived from a single ALL-patient sample (A0.1, A0.15, and A0.3 products). In the longitudinal analysis, transferred CAR⁻ T cells were not found to be significantly different from CAR⁺ T cells at the phenotypic level. The signaling provided during T-cell activation and manufacturing may overwhelm the effects of tonic or antigen-driven signaling from the CAR-construct[8]. Comparing the A0.15 and A0.3 products, significant differences were found in the potency and durability of antitumor response, and animal survival. Some of these observations could be explained by elevated expression of PD-1 and TIM-3 in CD8⁺ T cells in the A0.3 product compared to the A0.15 product. These differences manifested in fewer CAR-T cells being classified as highly cytotoxic in the A0.3 product in the in vitro analysis. Following infusion, we observed CAR-specific expansion in animals treated with the A0.3 product, suggesting that the cells were not dysfunctional, but possibly exhausted[16,29]. This finding is likely also a consequence of persistent antigenic stimulation from tumor cells or the lack of help from CD4⁺ T cells, which supports in vivo CD8⁺ CAR-T cell proliferation and differentiation[5,30,31]. Although A0.1 product-treated animals exhibited similar antitumor responses as those treated with the A0.15 product, we observed non-specific T-cell expansion (in 5/7 animals) which manifested in GVHD-like symptoms nearly 2 months after initial CAR-T cell dosing. The larger fraction of CD45RA⁺CCR7⁻ CD8⁺ effector CAR-T cells and highly cytotoxic CAR-T cells in this infusion product may explain these results. T cells in the A0.15 and A0.3 infusion products likely needed to proliferate and differentiate to

generate sufficient cytotoxic effector CD8[+] T cells to eliminate the tumor, depleting the store of memory T cells in the process. Whereas the A0.1 infusion product may have already contained enough effector T cells to clear the tumor. The remaining memory T cells, likely driven by signaling provided during initial T-cell activation, continuously expanded non-specifically, causing increased numbers of CAR-negative T cells and GVHD. While such an infusion product may not be safe in the context of B-cell malignancies, it could be advantageous in a solid tumor context, where CAR-T cells are required to rapidly debulk large portions of a tumor. While we observed similar trends in the stimulation dose dependency of CAR-T cell responsiveness in cells generated from a healthy donor sample, donor-to-donor variability should be an important consideration as the features of the T-cell sample directly impacts how the cells respond to stimulation. This further emphasizes the need for personalized manufacturing pipelines.

In our proof-of-concept analysis to quantify and predict the relationship between a T-cell blood sample and CAR-T cell product phenotype, stimulation dose was identified as the single most important parameter in defining CAR-T cell products regardless of health status. Heightened expression of T-cell exhaustion markers, including TIM-3 and LAG-3, in the CAR-T cell infusion[8] and apheresis products have been previously reported to be correlated with suboptimal disease outcomes[6,7]. Other features, such as the frequency of CD45RA[+]CCR7[−] CD8[+] T cells in the PBMC sample, were also determined to be important for patient-derived CAR-T cells, consistent with differences observed in our initial blood sample phenotyping. Higher sample numbers would likely improve the accuracy of our prediction model and allow it to describe−given an input T-cell blood sample phenotype −what CAR-T cell products are possible (and not possible) using APC-ms across of a range of stimulation dose. While we focus on 41BB/CD3ζ CARs, we expect an analogous approach could model CD28/CD3ζ CARs as well; however, given their distinctions in CAR signaling, we expect differences in the range of possible CAR-T cell product phenotypes. We envision that further refinement of our ML model, and its future iterations could be used as a tool for clinicians and manufacturers to personalize and guide the manufacture of more consistent and more functional autologous CAR-T cells for therapy.

# Methods

## Samples
Healthy de-identified blood collars were obtained from Brigham's and Woman's Hospital or Hemacare and processed in a Ficoll gradient (Sigma), washed several times in 1× PBS to remove platelets to obtain enriched peripheral blood mononuclear cells (PBMC) and frozen prior to use (Bambanker, Wako). Acute lymphoblastic leukemia (ALL) and chronic lymphocytic lymphoma (CLL) patient PBMCs were obtained from patients treated at the Dana-Farber Cancer Institute (DFCI) under Institutional Review Board (IRB)-approved protocols from patients who provided informed consent. Patients were male or female and varied in age. Handling and research involving these samples complied with all DFCI IRB ethical regulations. All healthy and patient-derived (i.e., ALL/CLL) PBMCs were phenotyped via flow cytometry prior to CAR-T cell production.

## Chemicals and reagents
Chemical reagents for mesoporous silica rod (MSR) synthesis were obtained from Millipore-Sigma (e.g., concentrated hydrochloric acid; Pluronic P123, Mn ~5800; tetraethyl orthosilicate, TEOS, 98%). 16:0–18:1 PC (POPC) and 18:1 biotinyl cap PE were obtained from Avanti Polar Lipids. Cell culture-related reagents were obtained from Lonza (e.g., RPMI 1640, X-VIVO 15), ThermoFisher (e.g., HI-FBS, EDTA), Gibco (e.g., penicillin–streptomycin), MilliporeSigma (e.g., bovine serum albumin; BSA), and GeminiBio (e.g., Human serum AB). Recombinant human IL-2 was obtained from Akron Biotech and titrated before use.

Streptavidin was obtained from VWR. Ligands for T cell activation were obtained from Biolegend (i.e., biotinylated anti-CD3, anti-CD28). Human CD3/CD28 T-cell expansion Dynabeads were obtained from ThermoFisher. Fixable blue dead stain was obtained from Thermo-Fisher. Anti-human antibodies for flow cytometry were obtained from BioLegend: CD3-PerCP/Cy5.5 (HIT3a), CD4-BV510 (SK3), CD8-APC/Fire750 (SK1), PD-1-PE (EH12.2H7), TIM-3-BV421 (F38-2E2), CD25-BV711 (M-A251), CD45RA-PE/Cy7 (HI100), CCR7-APC (G043H7), truncated EGFR-AF488 (AY13), CD95-PE/Dazzle (DX2), CD137-PE/Cy5 (4B4-1), CD3-PE/Dazzle (HIT3a), and CD25-PE/Cy5 (M-A251). Anti-mouse CD45-PerCP/Cy5.5 (30-F11) and Ly6G-BV711 (1A8) were obtained from Bio-Legend. All antibodies were used at the manufacturer-recommended dilution.

## APC-ms synthesis, assembly, and characterization
A step-by-step protocol for the synthesis, preparation, and assembly of APC-ms is available online[14].

**Mesoporous silica rod (MSR) synthesis.** High-aspect ratio MSRs were synthesized via the sol−gel method by dissolving 4 g Pluronic P123 in 150 g of 0.6 M HCl solution and stirred with 8.6 g of tetra-ethylorthosilicate at 40 °C for at 20 h. The suspension was transferred to a 100 °C oven and aged for 24 h. The synthesized MSRs were sieved using a 63 µm test sieve (VWR) and calcined at 550 °C for 4 h to remove leftover pluronic, contaminants, and endotoxin.

**Material characterization.** All MSR materials were analyzed before in vitro T-cell activation. Thermogravimetric analysis (TGA) was conducted to confirm removal of contaminants and the individual MSRs were processed for sizing via bright field microscopy (EVOS FL Cell Imaging System) and scanning electron microscopy (SEM) to determine rod dimensions and ultrastructure. Nitrogen physisorption (3Flex) was performed to determine the pore size, volume, and surface area of the MSRs. These analyses are summarized in Supplementary Table 2. MSRs were also evaluated for their ability to coat liposomes and release IL-2, relevant biological proxies to confirm consistent physical structure between batches. MSRs were further confirmed to be endotoxin free via the Limulus Amebocyte Lysate test (Charles River).

**APC-ms preparation and assembly.** Liposomes were prepared as previously described. Lipid films consisting of 1-palmitoyl-2-oleoyl-sn-glycero-3-phosphocholine (POPC) doped with 0.02–0.3 mol% 1,2-di-(9Z-octadecenoyl)-sn-glycero-3-phosphoethanolamine-N-(cap biotinyl) (biotinyl-cap-PE) were prepared and dried under nitrogen. The films were rehydrated in 1× PBS, incubated for 1 h with periodic vortexing, and sized (Mini-Extruder, Avanti Polar Lipids). The resulting liposomes were ~120 nm in diameter and used within one week. APC-ms were prepared by mixing monodisperse liposomes with MSRs at a 1:4 w/w (lipid:MSR) ratio for 1 h at room temperature with gentle periodic mixing. The lipid-coated MSRs were washed and blocked using 0.25% w/v bovine serum albumin. Anti-CD3 (OKT3) and anti-CD28 (CD28.2; Biolegend) were pre-mixed at a 1:1 ratio in appropriate quantities. Prior to the addition of the anti-CD3/anti-CD28 cocktail, streptavidin was added at a 30% theoretical saturation of the biotinyl-cap PE, and mixed periodically for 7 min. The assembled APC-ms were washed in 1× PBS and resuspended in X-VIVO 15 supplemented with 10% FBS and 1% P/S prior to T-cell activation and CAR-T cell production. Supplementary Table 3 outlines the APC-ms formulations used in this study.

## CAR-T cell production
**Lentivirus construction and production.** The second-generation CD19-41BBζ CAR construct was composed of the scFv fragment from the FMC63 antibody (GenBank: ADM64594.1) fused to the human

CD8α hinge and transmembrane region (Gene bank number NP_001759.3, aa 138–206) and linked to human 4-1BB (Gene bank number NP_001552.2, aa 214–255) and human CD3ζ (Gene bank number NP_000725, aa 52-163) intracellular signaling domains. To enable detection by flow cytometry, a cleavable truncated EGFR (tEGFR) was inserted to the N-terminus of the CD3ζ. Lentiviral supernatants were produced using the HEK 293T packaging line as previously described[32]. Briefly, 70% confluent HEK 293T cells in a well of a 6-well plate was co-transfected with 0.2 µg CAR-vector plasmid, 0.9 µg pMD2.G, 1.9 µg psPAX2 using lipofectamine 2000 (Life Technologies). The cultures were grown for 60 h, after which the supernatants were collected, filtered to remove debris, and frozen at −80 °C before use.

**Primary T-cell isolation, transduction, and expansion.** T cells were isolated from either healthy and patient samples using the human pan-T cell isolation kit (Miltenyi Biotec) to obtain untouched CD3⁺ T cells for CAR-T cell production. Isolated T cells were co-cultured with various APC-ms formulations (Supplementary Table 3) or Dynabeads (either 1:1, 3:1, or 5:1 bead:cell ratio) at a density of $3.13 \times 10^5$ cells/cm² in X-VIVO 15 supplemented with 10% FBS and 1% P/S. APC-ms was seeded at a material density of 93.75 µg/cm² (e.g., 30 µg in a 96-well plate with $1 \times 10^5$ isolated T cells). After 48 h, the activated T cells were transduced by adding 140 µL of pre-warmed lentiviral supernatant containing the CD19 CAR construct. After 36 h, the media containing T cells and any remaining material were transferred to a 6-well G-Rex plate (Wilson Wolf) containing pre-warmed X-VIVO 15 supplemented with 5% human AB serum (HABS) and 100 IU/mL IL-2 (Akron Biotech) and expanded for 5 days (total 8-day culture), then cryopreserved using Bambanker (Wako/Fujifilm). APC-ms were not pre-loaded with IL-2. The T-cell concentration was maintained between 0.1 and $2 \times 10^6$ cells/mL by adding fresh media containing 100 IU/mL IL-2 throughout the duration of culture. For Dynabead cultures, the beads were magnetically separated prior to cryopreservation. One CLL sample (CLL9) failed to proliferate following T-cell activation regardless of stimulation type (i.e., APC-ms or Dynabeads). The CAR transduction efficiency was consistently ~30–50%. An ELISA was used to measure the pg amount of surface anti-CD3 and anti-CD28 antibody presented on the beads or scaffolds to determine that the Dynabead D3:1 condition to be dose-matched with the APC-ms A0.1 condition.

## T-cell analysis

Following CAR-T cell production, cells were enumerated using an automatic cell counter (MUSE). Fold expansion was calculated by dividing the number of live cells by the number of seeded cells at the beginning of culture.

**Phenotyping.** At indicated time points, blood samples or CAR-T cell products were processed for flow cytometry. All anti-human antibodies were used at the manufacturer's recommended concentration. Flow cytometry using a BD LSRFortessa instrument. Gates were set using fluorescence minus one (FMO) controls. Laser intensities were standardized using Rainbow beads (Biolegend) to minimize variability between runs. Data was analyzed using FlowJo v10 (TreeStar). A gating strategy for T-cell blood samples is shown in Supplementary Fig. 1a and the gating strategy for CAR-T cell products is shown in Supplementary Fig. 1b. All CAR-T cell products were phenotyped to determine the frequency of CAR⁺ cells prior to the following analyses.

**Intracellular cytokine staining studies.** CD19-expressing Raji cells (ATCC) were luciferized and positively-selected using puromycin antibiotic selection. Luciferized Raji cells (Raji-luc) cells were used within 6 passages. Raji-luc cells were washed and were co-cultured with CAR-T cell products at a 5:1, 2:1, 1:1, 1:2, or 1:5 effector:target (CAR⁺ T-cell:Raji-luc) ratio in 96-well U-bottom plates (VWR) in a total volume of 100 µL RPMI supplemented with 10% HI-FBS. The absolute number of CAR⁺ T cells was fixed at $5 \times 10^4$. After 1 h of co-culture, Brefeldin A (BD Biosciences) was added to inhibit cytokine secretion, and the cells were cultured for an additional 3 h prior to flow cytometry analysis to evaluate the expression of intracellular cytokines (i.e., granzyme B, IFN-γ, TNF-α, and IL-2).

**Cytotoxicity studies.** Raji-luc target cells were pelleted with CAR-T cell products in 96-well U bottom plates (VWR) at effector:target (CAR⁺ T-cell:Raji-luc) ratios of 20:1, 10:1, 5:1, 2.5:1, 1.25:1, and 0:1 for 20 h. 1% triton X-100 (VWR) was used as a positive control. $2 \times 10^5$ CAR⁺ T cells and $1 \times 10^4$ target cells were used as the highest effector-to-target ratio in a total volume of 100 µL RPMI supplemented with 10% HI-FBS. After 20 h, CAR-T cells and Raji-luc co-cultures were lysed using Bright-GLO reagent (Promega) and the luminescence signal of the solution was quantified using a plate reader (Biotek H1). Percent cytotoxicity was calculated as $1 - (\text{sample} - \text{ctrl}_{\text{triton X-100}})/(\text{ctrl}_{0:1} - \text{ctrl}_{\text{triton X-100}})\%$.

## Xenograft lymphoma model

Female, NOD.Cg-Prkdcscid Il2rgtm1Wjl/SzJ mice (NSG) mice, between 6 and 7 weeks of age and ~20 g in weight (Jackson Laboratories), were used for in vivo therapeutic studies. Animals were maintained on 10–12 h light cycles at ambient temperature and humidity, and fed chow and water ad libitum. Procedures were approved by Harvard University's Institutional Animal Care and Use Committee and in compliance with National Institutes of Health guidelines.

**Tumor inoculation and CAR-T cell treatment.** NSG mice were inoculated with a high dose of $5 \times 10^5$ luciferized Raji cells (Raji-luc) intravenously on day 0. After 4 days, tumor-bearing mice were randomized into treatment groups ($n = 7$ each, sample size calculated by G*Power a priori analysis) and were treated with either mock (RPMI-1640) or $5 \times 10^5$ CAR⁺ T cells made using APC-ms presenting anti-CD3 and anti-CD28 at a 0.1, 0.15, or 0.3 mol% biotin stimulation dose (abbreviated A0.1, A0.15, and A0.3 products). Prior to CAR-T cell infusion, cryopreserved CAR-T cells were thawed in pre-warmed X-VIVO 15 supplemented with 5% human AB serum and enumerated via hemocytometer with trypan blue exclusion. Infusion products were washed at least three times in serum-free RPMI-1640 to remove serum and resuspended at the appropriate concentration for cell infusion. No additional purification steps were required.

**Tumor tracking.** Raji-luc tumor burden was monitored over time using D-Luciferin (Gold Biotechnology). Animals were anesthetized and intraperitoneally injected with D-Luciferin at 150 mg/kg. Luminescence was measured 10 min post injection via IVIS (Perkin Elmer). Total flux (p/s) per mouse was quantified in mouse whole-body regions of interest (ROI). Animals were imaged once every 4–14 days and their weights were simultaneously quantified. Mice were monitored daily for signs of discomfort and euthanized upon the development of hind-limb paralysis, when immediate and significant graft-versus-host disease (hair loss, behavior changes, health decline) was observed, or when more than 25% of the initial body weight was lost.

**Longitudinal CAR-T cell analysis.** At the indicated time points, animals were bled via the tail vein and ~50 µL blood was collected in K2-EDTA-coated collection tubes (BD). The samples were treated with ACK lysis buffer (Lonza), washed, and processed for flow cytometry as described above (BD LSRFortessa). A combination of anti-human and anti-mouse antibodies were used. A sample gating strategy is outlined in Supplementary Fig. 1c.

## Computational modeling
**Cytotoxicity and phenotype correlation (Fig. 2).** To correlate in vitro Raji-luc cell cytotoxicity with CAR T-cell product phenotype, single T-cell flow cytometry intensities were exported from FlowJo and

analyzed in R. The following analysis was performed separately for healthy donor or patient-derived samples. Each T-cell event (or single cell, as captured by flow cytometry) from a specific CAR-T cell product was assigned an experimentally observed % cytotoxicity value. To identify phenotypes associated with greater cytotoxic potential, $K$-means clustering was performed iteratively for a series of $k$-clusters from $k = 2$ to $k = 320$. For each $k$-cluster, the characteristic % cytotoxicity value for each cluster was calculated by averaging the experimentally assigned cytotoxicity of the T cells belonging to that cluster, hypothesizing that T cells with similar phenotypes will have similar killing potential. The optimal number of clusters was calculated based on an explained variance metric. First, the average feature expression per cluster for all the $k$-clusters was estimated. Next, a random forest regression was performed using the cluster features as the input and the average % cytotoxicity value of the $k$-cluster as the output. The optimal number of $k$-clusters was determined to have the highest explained variance by iterating $k$ from 2 to 320. To classify T-cell phenotype as low, medium, and highly cytotoxic potential, the average % cytotoxicity value was $z$-scored per cluster for the optimal number of $k$-clusters. Highly cytotoxic products were defined as clusters with $z$-scored cytotoxicity $\geq 1$, low cytotoxic products $\leq -1$ and medium cytotoxicity products between $-1$ and 1.

**Stimulation classification model (Fig. 4).** Separate classification models were developed for healthy donor and patient-derived samples. Flow cytometry data was gated according to Supplementary Fig. 1a, b to determine T-cell subpopulation features (e.g., %CD4$^+$, % CD8$^+$, etc.) for each healthy donor or patient sample prior to and after T-cell activation with APC-ms. All post-stimulation CAR-T cell product phenotypes were pre-processed by $z$-scoring each feature, represented by the various gated T-cell subpopulations. The pre-stimulation T-cell samples were pre-processed in the same manner. There are a total of 49 samples from ALL/CLL patient samples and 56 samples from healthy donors. To determine the relationship between CAR-T cell phenotype and the initial T-cell phenotype and stimulation dose, we assigned an APC-ms stimulation dose to each pre-stimulation T-cell sample and clustered similar CAR-T cell products via $K$-means clustering into five broad phenotypes of possible CAR-T cell products (using $k = 5$) to improve generalizability. Then, we assessed several ML classifiers (i.e., Random Forest classifier, Multi-Layer Perceptron classifier, Decision Tree classifier, Naive Bayes classifier, and Nearest Neighbors classifier, all obtained from the Python sci-kit learn 1.0.2 package) for their ability to classify CAR-T cell $k$-cluster phenotype based on the initial T-cell blood sample phenotype and APC-ms stimulation dose. Each model was tenfold cross-validated. In each repetition, 70% of the data set was used for training and cross-validating the model and 30% was held out for testing. For the training set, a total of 35 patient samples and a total of 40 healthy donor samples were used. When classifying the CAR-T cell cluster from stimulation dose and input T-cell phenotype, the model chose the samples randomly and across all phenotypes. No restriction was performed. A grid search was performed to tune the parameters for each model (e.g., number of estimators, maximum depth, number of features to consider, etc.), and the best parameters were used to assess the performance of the model on the test set. The tuned parameters that were used for the random forest model were 800 estimators, Gini impurity criterion, minimum of 2 samples at a leaf node, minimum of 2 samples to split an internal node, and a maximum depth of 10. For the random forest classifier, important features contributing to the model were assessed.

**Stimulation prediction model (Fig. 4).** To predict APC-ms stimulation dose directly based on a desired CAR-T cell phenotype and an initial T-cell blood sample phenotype, a nonlinear, random forest regression ML model was used. Similar to the classification model, flow cytometry

data was gated based on Supplementary Fig. 1a, b to determine T-cell subpopulation features of the initial T-cell blood sample and the CAR-T cell products, respectively. These were used as inputs in the model to predict the pg/cell level of APC-ms stimulation. Regression models were parameter-tuned and cross-validated on all donors besides three healthy donor samples, using the same 70/30 split. Then, to measure model robustness, the model was tested by training on samples from all donors except one, and then predicting the stimulation for a given sample using the CAR-T cell and initial T-cell blood sample phenotypes. Unlike the classification model where no cells were restricted to training, validation, or testing, in this prediction model, we restricted all cells from a particular donor to either training, validation, or testing when predicting the stimulation dose. This enables assessment of generalization to different donors. The predicted score was given by the coefficient of determination or $R^2$ between the predicted and actual stimulation values.

### Statistical analysis
Statistical analyses were performed using Prism v8.01 (Graphpad) and statistical software R v4.1.1 and Python 3.7. Equality of variances was determined using Levene's test. Normally distributed data were expressed as mean ± s.e.m. unless indicated otherwise. Data with non-normal distributions were expressed as the median ± interquartile range. In general, for samples of equal variance, appropriate parametric tests (e.g., paired Student's $t$ test) were used. For samples of unequal variance, appropriate non-parametric tests (e.g., Wilcoxon signed-rank test) were used. Significance of Kaplan–Meier survival curves were determined using log-rank Mantel–Cox tests. All tests were two-sided. The level of significance was set at $p < 0.05$. A trend was represented as $0.05 < p < 0.1$. Detailed statistical methods and $p$ values are reported in the figures and figure captions. A table reporting $p$ values is shown in Supplementary Table 5.

### Reporting summary
Further information on research design is available in the Nature Portfolio Reporting Summary linked to this article.

## Data availability
Patient-related data that were collected but not shown in the paper might be subject to confidentiality (e.g., sex). Data supporting the findings of the study are available within the main manuscript or Supplementary Information. All outstanding data requests should be made to the corresponding author. Source data are provided with this paper.

## Code availability
All computational analyses and data used for model training have been made available at: https://github.com/siddharthriyer/car_t_stimulation.

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

## Acknowledgements

This work was supported by the Wyss Institute at Harvard University (D.J.M.) and the Food and Drug Administration (5R01FD006589). Part of this work was supported by the National Cancer Institute of the National Institutes of Health under Award Number U54CA244726. The content is solely the responsibility of the authors and does not necessarily represent the official views of the National Institutes of Health. D.K.Y.Z. acknowledges support from the Canadian Institutes of Health Research (CIHR). Material characterization was performed at the Center for Nanoscale Systems (CNS) at Harvard University, which is supported by the National Science Foundation (NSF; 1541979). Patient samples were obtained through the DFCI Pasquarello Tissue Bank. We thank Dr. K. Vining, Dr. A. Cheung, and Dr. A. Li for insightful scientific discussions. Additionally, we are grateful to E. Zigon, G. Cuneo, T. Ferrante, and H. Ijaz for assistance with flow cytometry, imaging, and material characterization. Lastly, we are grateful to the patients who donated their cells, which allowed us to perform this study.

## Author contributions

D.K.Y.Z. and D.J.M. conceived the study. D.K.Y.Z., K.A.B., Y.L., and J.M.B. performed experiments. S.I. K.A.B., and D.K.Y.Z. developed the computational models and analyzed data. D.N. supervised the statistical analyses. N.C. and C.W. supervised patient sample procurement and study design. D.K.Y.Z., K.A.B., S.I., and D.J.M. wrote the manuscript. All authors read and approved the manuscript.

## Competing interests

D.J.M. is an inventor on patent applications Harvard University has filed in relation to the aAPC-ms technology. That intellectual property has been licensed to Lyell Immunopharma, and D.J.M. has stock in Lyell Immunopharma. C.J.W. holds equity in BioNTech. The remaining authors declare no competing interests.
