## [Peer Review File · Nature Communications]

Enhancing CAR-T cell functionality in a patient-specific mannerREVIEWER COMMENTS

Reviewer #1 (Remarks to the Author):

The manuscript by Zhang et al describe the use of an artificial T cell stimulation technology that enables titration of signal strength to define patient-specific activation parameters to generate a CAR T cell product with optimized functionality. The data demonstrate the donor to donor variability that has been observed in both CAR T cell development work and in clinical experience with more precise measurement of this variability. T cells for cancer patients (CLL and ALL) performed differently than healthy donors as has been reported previously. The methodology is sound and the data has the potential to rationally optimize product manufacturing to generate CAR T cell products with the highest functionality. The data indicates that the approach described can be used to define optimized manufacturing conditions for both cancer patients and healthy donors and is, thus, applicable to both autologous and allogeneic products.

Comments:

1. The in vivo correlative data in figure 3 is important as this type of data has been most informative in understanding potency in pre-clinical models. This figure includes data from a single patient. Given donor to donor variability well-known to the field and nicely demonstrated by the authors, expanding in vivo potency testing to additional donor products generated using ms APC.
2. Given that the donor to donor variability and differences between healthy donor vs patient-derived material has been well described in the literature, the most important output is the predictive algorithm to identify optimal signal strength for a given T cell source based on attributes that can be measured in starting material. What is not shown in the manuscript is how well this algorithm performs using donor material not used for the development of the algorithm.
3. A limitation of the data is the use of a single CAR construct. Inclusion of another CAR (for example, one with different costimulatory domain) would enhance the broader applicability of the approach.

Reviewer #2 (Remarks to the Author):

The study of Zhang and colleagues presents a model to manufacture CD19 CAR Tc with enhanced functions. The relationship between quality and efficacy of a CAR-by flow cytometry assessed clinical product has been reported by others, but in the present article the authors support their finding using experimental models. The production of CAR T cells is a challenging process which lacks uniformity and predictability. Indeed, patient T cells might be of varying quality due to the treatment(s) they received. The authors propose to use a prediction model based on the quality of the stimulation of the T cells, a necessary step for transduction and expansion. Their study is performed using the blood samples of eight healthy donors and six patients. The authors also used well-defined markers and present a clear gating strategy from which they will extract their data. By comparing the effect of the dosing of two different products (Dynabeads and their synthetic APC (APC-ms)) on isolated T cells, they showed, using multiparametric analysis, that the cell profile and activity (cytotoxicity and cytokine release) of the T cells tend to change. As predicted, healthy donor T cells generated a more robust product. The authors then moved to an in vivo model and observed that the effects of the pre-treatment were obvious when T cells from one patient were stimulated with three formulations of APC-ms. Here, a dramatic difference in the outcome was observed, supporting the proposition that the quality of the T cell product can be influenced by the flavour of the stimulation. Finally, the authors propose a model linking T cell product quality (blood and CAR) to ideal stimulation dose, paving the road for future prediction tools which could assist clinician/production laboratories to personalize CAR Tc manufacturing for optimal therapeutic effect.

The reviewer captured the following main messages: there is a difference in expansion between healthy and patient T cells, but the differences in activity between the two batches when tested in vitro are not paramount. Yet, the impact of the patient treatment becomes obvious when the CAR

T cell product is tested in vivo. A few parameters detected by flow cytometry appeared sufficient to run the prediction model.

The topic is highly relevant and the experiments are well presented. However, the reviewer has one major concern regarding the in vivo validation experiment and a few minor points to be addressed:

Major:

- Fig. 3: The in vivo experiment is probably the most important demonstration of the impact of T cell stimulation on the treatment outcome, however, the design of the experiment is problematic. First, the figure legend should be more descriptive concerning the doses (numbers of animals), the treatments and the number of times the experiment was performed/repeated. Second, it is understood that the authors here want to illustrate the difference between the CAR Tc product from three different stimulations. However, the mock control is, according to the Methods section, medium alone. Although sometimes used, this procedure is debatable because the influence of the naked T cells (non-transduced or transduced with an irrelevant CAR) might affect the tumour growth. This is even more relevant in the present article because the authors report a differential GVHD effect that could also influence the tumour progression. It would therefore be more appropriate to use controls from T cells expanded with the same APC-ms and mock transduced. Third, although the authors explained why they focused on the APC-ms, a comparison with the reference product (Dynabeads) would be pertinent and supportive of the in vitro observations.

Minor:

- The title is misleading, in the present paper the authors are not "enhancing", but rather they are describing "A model to enhance...". Indeed, the in vivo data (which is in the reviewer's view the most striking demonstration of the strength of their model) has been performed using APC-ms. To show that the authors are "enhancing" they might have compared it with the Dynabeads standard formulation.

- Although well written, the topic is dense and more explanatory pictures could help to understand the importance of some experiments. For example, the Tc phenotype in Fig. 1 or Fig. 2F.

- Did the authors look for the presence of Treg in the leukapheresis product or after expansion? Could this influence the difference between patients and healthy donors. What is the Th1/Th2 status?

Reviewer #3 (Remarks to the Author):

Summary: Zhang et al. describe an analysis of CAR-T infusion products generated by APC-ms, which allows more refined control of antiCD3/antiCD28 stimulation dosage than the Dynabeads currently in use. They characterize CAR-T products generated under various levels of stimulation and build machine learning models to predict CAR_T product cytotoxicity and optimal APC-ms dosage based on flow-based features of isolated T cells. The paper is well written, the premise is exciting and the data and results support the authors conclusions. However, aspects of the computational methods are not described in sufficient detail.

Major comments:

The authors should provide more detail about the computational methods in the methods section. Specifically, what version of each ML model (Random Forest, FNN, DT, NB, NN) was used? What was the training set (number of examples and how distributed across phenotype)? What parameters were used? Did the authors perform any parameter optimization?

For each application of the classifiers, it is unclear what the number of training examples was, and whether the authors sought to control for shared substructure such as cells coming from the same donor. For example, are all T cells from a particular individual restricted to be either in the training or validation partition during cross validation? Line 582 seems to indicate that this was considered for the classifier to predict optimal APC-ms stimulation. Also, the authors say they performed 10-fold cross validation with a 70/30 holdout. Typically, one would split the training data into 70 / 30 train and test, then perform the 10-fold cross validation within the 70% for training, withholding

the 30% for estimation of generalization error at the end.

The authors try to identify the optimal k to distinguish groups of T cells with similar cytolytic activity using an iterative approach with k means clustering followed by Random Forest regression. Could the authors have simply used a linear regression or GAM rather than a Random Forest? Were there sufficient training examples and features to merit the Random Forest?

Figure 3 appears to be based on derivatives from 1 ALL sample, and it's unclear how consistent the observed effect sizes are likely to be. This should be mentioned as a caveat in the discussion.

For determining which dose generates the optimal CAR-T product (Figure 4), multiple machine learning tools are compared. Here, a simple decision tree performs as well as a random forest. Stimulation dose is by far the most important feature. The other features are not shown in Figure 4d and e. How well would stimulation dose do on its own? Perhaps it could be added as a baseline to 4c? This would help the reader understand how much additional information is gained by analyzing the characteristics of circulating T cells (i.e. how much the CAR-T prep can be personalized).

Data sharing: It is unclear why all of the data requires an MTA. Much of the data generated (e.g. flow cytometry values, % cytotoxicity assays, etc) are not identifiable and don't seem to even require controlled access. For example, feature values, CAR-T product labels and % cytotoxicity could easily be included in a supplementary excel sheet.

The authors should consider releasing their code and data analysis notebooks via Github or similar to support reproducibility of their work.

Minor comments:

Methods line 535: What do the authors mean by a 'T cell event' here? Does that just refer to a T cell from a specific CAR-T product? Is the idea that each T cell is characterized by a vector of features measured by flow cytometry? What features were used? Then T cells 'inherit' the % cytotoxicity of the CAR-T product. They are clustered based on the flow features, and the clusters are ascribed that average % cytotoxicity of the constituent T cells? The authors never actually mention the value of k in the ms, though from figure 2e it appears to be ~ 80 .

Figure 3A – both axes are labeled PC1

Figure 4 d and e, the features are not labeled

Line 81-82 is not a complete sentence.

Line 100: Tim-3 should be TIM-3

Line 309: Missing . after GVHD

Line 311: Did the authors mean portions rather than proportions?

Point-by-Point Response to Reviewer's Comments (Zhang et al. formerly manuscript NCOMMS-22-16280)

Please see below for our responses to each comment from every reviewer. The reviewer's comments are in regular typeface, and our replies in bold typeface.

Response to editor:

We are grateful to the editor and the editorial staff for their time and consideration in reviewing our manuscript. Thank you for the constructive feedback, which guided us in generating additional data to improve the strength and substance of our manuscript.

Reviewer #1 (Remarks to the Author):

The manuscript by Zhang et al describe the use of an artificial T cell stimulation technology that enables titration of signal strength to define patient-specific activation parameters to generate a CAR T cell product with optimized functionality. The data demonstrate the donor to donor variability that has been observed in both CAR T cell development work and in clinical experience with more precise measurement of this variability. T cells for cancer patients (CLL and ALL) performed differently than healthy donors as has been reported previously. The methodology is sound and the data has the potential to rationally optimize product manufacturing to generate CAR T cell products with the highest functionality. The data indicates that the approach described can be used to define optimized manufacturing conditions for both cancer patients and healthy donors and is, thus, applicable to both autologous and allogeneic products.

We thank the reviewer for their critical and insightful comments.

Comments:

1. The in vivo correlative data in figure 3 is important as this type of data has been most informative in understanding potency in pre-clinical models. This figure includes data from a single patient. Given donor to donor variability well-known to the field and nicely demonstrated by the authors, expanding in vivo potency testing to additional donor products generated using ms APC.

We agree on this important point. As suggested by the other reviewers, we have completed an additional series of studies using CAR-T cells derived from a separate donor. As requested by Reviewer #2, we have also included additional controls in this study, including CAR-T cells generated from Dynabeads and non-transduced T cells. CAR-T cells were generated using APC-ms presenting either 0.1 mol% or 0.3 mol% anti-CD3/anti-CD28 polyclonal stimulation (A0.1 and A0.3, respectively) and their therapeutic index were evaluated in the Raji lymphoma model. We selected a healthy donor sample, which unlike patient samples, are less sensitive to higher-dose stimulation (as described in Figs. 2 and 4), which we felt represented a higher and more difficult bar to demonstrate therapeutic differences because of stimulation dose. Importantly, to highlight the impact of stimulation on therapeutic response, we dosed tumor-bearing animals with either 5e5,

1e6, 1.5e6, or 3e6 CAR-T cells and observed that at least 3-6X more A0.3 CAR-T cells (i.e., 1.5-3e6) were needed to generate a similar therapeutic response as A0.1 CAR-T cells (i.e., 0.5e6).

We have included these results as new Supplementary Fig. 6 and have amended the main text with the following:

“Similar trends in CAR-T cell responsiveness were observed using A0.1 and A0.3 products generated from a separate healthy donor T-cell sample (Supplementary Fig. 6). Increasing the administered CAR-T cell dosage of the A0.3 product 3-fold, from 5×10^5 to 1.5×10^6 did not recover the therapeutic outcomes of the A0.1 product (Supplementary Fig. 6a-g). When comparing the A0.1 product and its dose-matched D3:1 product, we observed the A0.1 product showed improved in vitro cytotoxic function than their Dynabead counterparts (Supplementary Fig. 6h), which translated to improved animal survival in vivo (Supplementary Fig. 6i-j).” (lines 204-210).

2. Given that the donor to donor variability and differences between healthy donor vs patient-derived material has been well described in the literature, the most important output is the predictive algorithm to identify optimal signal strength for a given T cell source based on attributes that can be measured in starting material. What is not shown in the manuscript is how well this algorithm performs using donor material not used for the development of the algorithm.

Thank you for the comment. We agree this is an important point. To extrapolate the robustness of the model to new donors previously not used for model training, we trained the model on ALL/CLL samples, and then tested it on *healthy donor T-cell samples* and observed that the model was able to extrapolate modestly well to all healthy donors which were not present in the training/tuning data; $r^2 = 0.37$ (range: -0.43 to 0.78) for Random Forest. The accuracy varied from donor to donor, with negative scores for HD1 and HD2 (-0.43, -0.34, respectively) to $r^2 = 0.78$ and 0.75 for HD5 and HD8, respectively. This shows that the models could likely improve with additional data from more diverse donors. However, given that the r^2 is still fairly positive and that the model is able to predict r^2 for ALL/CLL donors robustly, we expect that our model will still be useful in the clinic, given that most CAR-T products are autologously-derived from cancer patients.

3. A limitation of the data is the use of a single CAR construct. Inclusion of another CAR (for example, one with different costimulatory domain) would enhance the broader applicability of the approach.

While we agree that it would be interesting to investigate another CAR construct, and in particular one with a different combination of intracellular costimulatory domains, such as CD28/CD3 ζ , our principal focus was to conduct a proof-of-concept demonstration of the effects of tuning stimulation strength during T-cell activation on resulting CAR-T cell phenotype and function. Previous studies have demonstrated that 2nd generation CD28/CD3 ζ CAR constructs can facilitate more rapid T-cell proliferation but are associated with poor persistence. Phosphoproteomic analyses have demonstrated that part of this may be explained by kinetic differences in CAR signaling - CD28/CD3 ζ signals

faster and in greater magnitude, whereas 41BB/CD3 ζ CAR T cells signals slower and drives memory-associated gene expression (Salter et al., 2018). Given the importance of signal strength in driving CAR-specific signaling events, we expect that stimulation dose is highly important in initiating such signaling events and therefore, analogous models to the ones we describe here would be applicable and relevant. The primary difference would be the range of possible CAR-T cell products - we expect some overlap with the 41BB/CD3 ζ CAR system we describe here, but certain phenotypes may not be achievable with CD28/CD3 ζ CARs.

We have added to the discussion section to touch on these points:

“While we focus on 41BB/CD3z CARs, we expect an analogous approach could model CD28/CD3z CARs as well; however, given their distinctions in CAR signaling, we expect differences in the range of possible CAR-T cell product phenotypes.” (lines 337-340)

Reviewer #2 (Remarks to the Author):

The study of Zhang and colleagues presents a model to manufacture CD19 CAR Tc with enhanced functions. The relationship between quality and efficacy of a CAR-by flow cytometry assessed clinical product has been reported by others, but in the present article the authors support their finding using experimental models. The production of CAR T cells is a challenging process which lacks uniformity and predictability. Indeed, patient T cells might be of varying quality due to the treatment(s) they received. The authors propose to use a prediction model based on the quality of the stimulation of the T cells, a necessary step for transduction and expansion. Their study is performed using the blood samples of eight healthy donors and six patients. The authors also used well-defined markers and present a clear gating strategy from which they will extract their data. By comparing the effect of the dosing of two different products (Dynabeads and their synthetic APC (APC-ms)) on isolated T cells, they showed, using multiparametric analysis, that the cell profile and activity (cytotoxicity and cytokine release) of the T cells tend to change. As predicted, healthy donor T cells generated a more robust product. The authors then moved to an in vivo model and observed that the effects of the pre-treatment were obvious when T cells from one patient were stimulated with three formulations of APC-ms. Here, a dramatic difference in the outcome was observed, supporting the proposition that the quality of the T cell product can be influenced by the flavour of the stimulation. Finally, the authors propose a model linking T cell product quality (blood and CAR) to ideal stimulation dose, paving the road for future prediction tools which could assist clinician/production laboratories to personalize CAR Tc manufacturing for optimal therapeutic effect.

The reviewer captured the following main messages: there is a difference in expansion between healthy and patient T cells, but the differences in activity between the two batches when tested in vitro are not paramount. Yet, the impact of the patient treatment becomes obvious when the CAR T cell product is tested in vivo. A few parameters detected by flow cytometry appeared sufficient to run the prediction model.

We thank this reviewer for their kind remarks and thoughtful comments which enabled us to strengthen the quality of the manuscript.

The topic is highly relevant and the experiments are well presented. However, the reviewer has one major concern regarding the in vivo validation experiment and a few minor points to be addressed:

Major:

- Fig. 3: The in vivo experiment is probably the most important demonstration of the impact of T cell stimulation on the treatment outcome, however, the design of the experiment is problematic. First, the figure legend should be more descriptive concerning the doses (numbers of animals), the treatments and the number of times the experiment was performed/repeated. Second, it is understood that the authors here want to illustrate the difference between the CAR Tc product from three different stimulations. However, the mock control is, according to the Methods section, medium alone. Although sometimes used, this procedure is debatable because the influence of the naked T cells (non-transduced or transduced with an irrelevant CAR) might affect the tumour growth. This is even more relevant in the present article because the authors report a differential GVHD effect that could also influence the tumour progression. It would therefore be more appropriate to use controls from T cells

expanded with the same APC-ms and mock transduced. Third, although the authors explained why they focused on the APC-ms, a comparison with the reference product (Dynabeads) would be pertinent and supportive of the *in vitro* observations.

We agree. We have modified the figure caption in Fig. 3 to include additional details regarding the dose of CAR-T cells administered, the number of animals used in each group, and experimental replicates:

“5 x 10⁵ luciferized Raji cells were administered four days prior to treatment with 5 10⁵ CAR+ T cells. CAR-T cell products generated via APC-ms from ALL10 was used for the animal study: mock, n=5; A0.1 (0.1 mol% APC-ms stimulated CAR-T cells), n=7; A0.15 (0.15 mol% APC-ms stimulated CAR-T cells), n=7; A0.3 (0.3 mol% APC-ms stimulated CAR-T cells), n=7 mice” (lines 713-717) and “Data in (k) represent mean ± s.e.m. and represents one experimental replicate.” (lines 722-724).

Thank you for the important comment regarding non-transduced T cells - this is an important point. We repeated our *in vivo* study and added a non-transduced T-cell control group. These T cells were generated following stimulation with APC-ms presenting 0.1 mol% anti-CD3/anti-CD28 and expanded for 8 days following the same protocols. No statistically significant differences were observed between non-transduced T cells and the mock control group (p>0.99). This new data is shown in Supplementary Fig. 6i.

Regarding the comparison to Dynabeads, in the same study, we treated tumor-bearing animals with CAR-T cells generated from APC-ms presenting 0.1 mol% or 0.3 mol% anti-CD3/anti-CD28 activating antibodies (termed A0.1 and A0.3 products, respectively) and Dynabeads presented at a 1:1 or 3:1 bead-to-cell ratio (termed D1:1 and D3:1 product, respectively). We observed that the A0.1 product significantly outperformed its dose-matched D3:1 product counterpart and improved animal survival (p=0.0019; Supplementary Fig. 6i). Although not significant, we observed a trend suggesting that the A0.1 product outperforms the D1:1 product as well (e.g., median survival 41 days vs 34 days, respectively). This new data is shown in new Supplementary Fig. 5 and we have modified the text to highlight these new findings.

“Similar trends in CAR-T cell responsiveness were observed using A0.1 and A0.3 products generated from a separate healthy donor T-cell sample (Supplementary Fig. 6). Increasing the administered CAR-T cell dosage of the A0.3 product 3-fold, from 5 x 10⁵ to 1.5 x 10⁶ did not recover the therapeutic outcomes of the A0.1 product (Supplementary Fig. 6a-g). When comparing the A0.1 product and its dose-matched D3:1 product, we observed the A0.1 product showed improved *in vitro* cytotoxic function than their Dynabead counterparts (Supplementary Fig. 6h), which translated to improved animal survival *in vivo* (Supplementary Fig. 6i-j).” (lines 204-210).

Minor:

- The title is misleading, in the present paper the authors are not “enhancing”, but rather they are describing “A model to enhance...”. Indeed, the *in vivo* data (which is in the reviewer’s view the most striking demonstration of the strength of their model) has been performed using APC-ms. To show that the authors are “enhancing” they might have compared it with the Dynabeads standard formulation.

Thank you for the important comment. To demonstrate that we are enhancing CAR-T cell functionalities, we included the Dynabead-stimulated CAR-T cell reference products as requested. In addition to the dose-matched D3:1 condition (compared to A0.1), we included a D1:1 CAR-T cell product. As mentioned in the above comment, we observed that the A0.1 CAR-T cell product outperformed its dose-matched D3:1 CAR-T cell product counterpart, highlighting the enhancement in CAR-T cell functionality. This data is shown as new Supplementary Fig 6h-j, and we have amended the text accordingly. These data are in line with our functional analysis of Dynabead-generated and APC-ms generated CAR-T cell products in Fig. 2.

- Although well written, the topic is dense and more explanatory pictures could help to understand the importance of some experiments. For example, the Tc phenotype in Fig. 1 or Fig. 2F.

Thank you for the comment. We have modified Fig 1i and Fig. 2f to improve the clarity of the key takeaways. In Fig. 1i, we have added an arrow to highlight the differentiation status of the CAR-T cells (i.e., from memory to effector T cells). In Fig. 2f, we have reorganized the T-cell markers and added labels to classify them as either T-cell activation/exhaustion markers or T-cell memory markers. Please let us know if there's anything else we can do here to improve the clarity of the figures.

- Did the authors look for the presence of Treg in the leukapheresis product or after expansion? Could this influence the difference between patients and healthy donors. What is the Th1/Th2 status?

This is an interesting comment. To understand the impact of Th1/Th2/Treg status in the functional difference between healthy and patient-derived CAR-T cells, we performed a deeper phenotypic FACS analysis. We focused on the Dynabead 1:1 (bead-to-cell ratio)-stimulated CAR-T cell products due to their clinical relevance. In Fig. 2d, we highlighted differences between healthy and patient-derived D1:1 CAR-T cells and showed that patient-derived D1:1 CAR-T cells were less cytotoxic than their healthy donor counterparts. We found no significant differences in Foxp3 expression between healthy and patient-derived D1:1 CAR-T cells (Response Fig. 1a). Using CCR6 and CXCR3 to distinguish Th1/Th2/Th17 cells, we observed no significant differences between healthy and patient-derived D1:1 CAR-T cells (Response Fig. 1b). We did observe a trend suggesting more CCR6⁺CXCR3⁺ CD4⁺ T cells in healthy donors than patient samples. However, when we polyclonal-stimulated the CAR-T cells using PMA/Ionomycin, we observed largely similar Th responses and no significant difference in the Th1/Th2 ratio, as measured by dividing the % of IFN- γ ⁺ CD4⁺ T cells by the % of IL-4⁺ CD4⁺T cells (Response Fig. 1c). A minor trend was observed with the patient D1:1 product expressing more IFN- γ than healthy D1:1 products. From our previous analysis, we observed more IL-4⁺ CD4⁺ CAR-T cells among healthy donor CAR-T cells than patient-derived CAR-T cells (Response Fig. 4d). These data suggest that the contribution of Tregs and functional Th responses to the therapeutic index is likely minimal when compared to CD4-derived IL-2.

We have amended the Results section: “Additionally, similar to the trends observed in CD8⁺ CAR-T cells, healthy CD4⁺ CAR-T cells expressed more IL-2 than patient-derived CD4⁺ CAR-T cells (Supplementary Fig. 4e).” (lines 128-129)

Response Fig. 1. (a) FcγR3 expression among healthy and patient-derived Dynabead 1:1 (bead: cell ratio)-stimulated CAR-T cell products (D1:1 product). (b) Expression of CCR6 and CXCR3 among healthy and patient-derived D1:1 products. (c) Th1/Th2 ratio of healthy and patient-derived D1:1 products. (d) IL-2 expression in CD4⁺ CAR-T cells from healthy and patient-derived samples following Raji cell stimulation. Data represents median ± interquartile range. Data in (a-c) represents n=5 for healthy D1:1 products and n=5 for patient D1:1 products. Data in (d) represents n=4 healthy or patient-derived CAR-T cell products for each stimulation condition. Th1/Th2 ratio in (c) was calculated by dividing the frequency of IFN-γ⁺ CD4⁺ T cells by the frequency of IL-4⁺ CD4⁺ T cells.

Reviewer #3 (Remarks to the Author):

Summary: Zhang et al. describe an analysis of CAR-T infusion products generated by APC-ms, which allows more refined control of antiCD3/antiCD28 stimulation dosage than the Dynabeads currently in use. They characterize CAR-T products generated under various levels of stimulation and build machine learning models to predict CAR_T product cytotoxicity and optimal APC-ms dosage based on flow-based features of isolated T cells. The paper is well written, the premise is exciting and the data and results support the authors conclusions. However, aspects of the computational methods are not described in sufficient detail.

We are grateful to this reviewer for their comments and expertise in the ML space. Their feedback allowed us to greatly improve the clarity, accuracy, and reproducibility of the manuscript.

Major comments:

The authors should provide more detail about the computational methods in the methods section. Specifically, what version of each ML model (Random Forest, FNN, DT, NB, NN) was used? What was the training set (number of examples and how distributed across phenotype)? What parameters were used? Did the authors perform any parameter optimization?

We have now provided additional detail regarding the models that were used. These details were included in the *stimulation classification model section* of the methods (Lines 591-628), including details regarding the version of the models:

“Random Forest classifier, Multi-Layer Perceptron classifier, Decision Tree classifier, Naïve Bayes classifier, and Nearest Neighbors classifier, all obtained from the Python sci-kit learn 1.0.2 package” (line 578-580).

For the training set, the number of samples used included 35 ALL/CLL patient samples and 40 healthy donor samples. When attempting to classify the final product k-cluster from stimulation dose and input T-cell phenotype, the model chose the samples randomly and evenly across all phenotypes. When predicting the optimal stimulation given a desired CAR-T cell product phenotype and input blood sample phenotype, the entirety of all CAR-T cell product samples from one donor were held out to evaluate if the model could generalize to a donor it had never seen before.

The tuned parameters that were used for the random forest model were 800 estimators, Gini impurity criterion, minimum of 2 samples at a leaf node, minimum of 2 samples to split an internal node, and a maximum depth of 10. These parameters were optimized only on the training set, and then the best parameters were used to assess model performance on the test set. We have modified the text accordingly:

“For the training set, a total of 35 patient samples and a total of 40 healthy donor samples were used. When classifying the CAR-T cell cluster from stimulation dose and input T-cell phenotype, the model chose the samples randomly and across all phenotypes. No restriction was performed. A grid search was performed to tune the parameters for each model (e.g., number of estimators, maximum depth, number of features to consider, etc.), and the best parameters were used to assess the performance

of the model on the test set. The tuned parameters that were used for the random forest model were 800 estimators, Gini impurity criterion, minimum of 2 samples at a leaf node, minimum of 2 samples to split an internal node, and a maximum depth of 10.” (lines 583-590)

For each application of the classifiers, it is unclear what the number of training examples was, and whether the authors sought to control for shared substructure such as cells coming from the same donor. For example, are all T cells from a particular individual restricted to be either in the training or validation partition during cross validation? Line 582 seems to indicate that this was considered for the classifier to predict optimal APC-ms stimulation. Also, the authors say they performed 10-fold cross validation with a 70/30 holdout. Typically, one would split the training data into 70 / 30 train and test, then perform the 10-fold cross validation within the 70% for training, withholding the 30% for estimation of generalization error at the end.

Thank you for the important comment. Our apologies for the confusion. That is correct - 30% of the samples we split off before the cross-validation and parameter tuning. Then, the model accuracy was assessed on these 30% which are used as a test set.

Regarding restricting T cells from a particular individual, we did restrict all the cells from a given donor to either training, validation, or testing when predicting the optimal stimulation dose. This was done so that we could make a usable model for clinical samples, enabling generalization to different donors.

However, for the other classifiers we present (differentiating healthy from patient samples or for predicting output CAR-T cell phenotype cluster) we did not have this restriction as the purpose of those models was simply for feature selection / interpretation.

We have modified the text in the method section to clarify this point:

“Regression models were parameter-tuned and cross-validated on all donors besides three healthy donor samples, using the same 70/30 split. Then, to measure model robustness, the model was tested by training on samples from all donors except one, and then predicting the stimulation for a given sample using the CAR-T cell and initial T-cell blood sample phenotypes. Unlike the classification model where no cells were restricted to training, validation, or testing, in this prediction model, we restricted all cells from a particular donor to either training, validation, or testing when prediction the stimulation dose. This enables assessment of generalization to different donors.” (lines 597-603)

The authors try to identify the optimal k to distinguish groups of T cells with similar cytolytic activity using an iterative approach with k means clustering followed by Random Forest regression. Could the authors have simply used a linear regression or GAM rather than a Random Forest? Were there sufficient training examples and features to merit the Random Forrest?

We appreciate this reviewer’s comments regarding alternative regression approaches. We selected Random Forest mainly because linear regression and GAM assume linearity

in the data. While this could be true for our specific dataset, we decided to choose a model that does not require this assumption. However, we acknowledge that it is advisable to compare multiple regression models for any given dataset. Consequently we applied GAM models, and saw the similar results (Response Fig. 2a).

We also acknowledge the concern regarding the size of the datasets. The size of the data will be directly correlated with the number of iterative clusters generated. Note that the k-clusters are generated directly from single-cell flow cytometry intensity values. Consequently, it is expected that smaller k-clusters will result in poor Random Forest regression and low R^2 values. We expect an increase in the R^2 values as k increases, until a point when a further increase in k will have minimal impact on the R^2 value (see Response Fig. 2b). It is at this point where we define our optimum number of k-clusters.

Response Fig. 2. (a) Comparison of Random Forest and GAM regression for cytotoxic potential classification. (b) R^2 as a function of increasing k using GAM regression. Similarly high R^2 values were observed when GAM models were performed with higher numbers of k -clusters.

We have added this comparison as Supplementary Fig. 4h-i. We have further added to the main text:

“We chose Random Forest regression over linear regression or GAM since the latter two assume linearity. GAM models showed similar results (Supplementary Fig. 4h, i).” (lines 152-154).

Figure 3 appears to be based on derivatives from 1 ALL sample, and it’s unclear how consistent the observed effect sizes are likely to be. This should be mentioned as a caveat in the discussion.

This is an important note. As suggested by the other reviewers, we repeated our studies in Figure 3 to include CAR-T cells derived from a healthy donor, which we felt

represented a higher bar demonstration of the stimulation dose dependency of CAR-T cell responsiveness, as healthy donor T cells appeared less sensitive to stimulation than patient-derived T cells (Fig. 2). As expected, we observed a similar stimulation dose dependency in CAR-T cell responsiveness, demonstrating that trends are consistent across donors. However, it is difficult to pinpoint the precise observed effect sizes, as they would be subject to donor-to-donor variability, so we have added a caveat to the discussion as suggested:

“While we observed similar trends in the stimulation dose dependency of CAR-T cell responsiveness in cells generated from a healthy donor sample, donor-to-donor variability should be an important consideration as the features of the T-cell sample directly impacts how the cells respond to stimulation. This further emphasizes the need for personalized manufacturing pipelines.” (lines 338-342).

For determining which dose generates the optimal CAR-T product (Figure 4), multiple machine learning tools are compared. Here, a simple decision tree performs as well as a random forest. Stimulation dose is by far the most important feature. The other features are not shown in Figure 4d and e. How well would stimulation dose do on it’s own? Perhaps it could be added as a baseline to 4c? This would help the reader understand how much additional information is gained by analyzing the characteristics of circulating T cells (i.e. how much the CAR-T prep can be personalized).

Thank you for the comment and important note. Both the MLPC and decision tree perform about as well as the random forest model, but we chose random forest models to investigate and proceed with as they were more consistent than decision trees and more interpretable.

Using our best random-forest model, stimulation dose by itself achieves prediction accuracies of ~30-40% on the test sets for patient-derived and healthy donor samples. When all the T-cell features are available, accuracy improves to 80% and 70% respectively for patient and healthy samples. This data is shown in new Supplementary Table. 4:

With Features	Model type	Random forest	MLPC	Decision Tree	Nearest Neighbors	Naïve Bayes
	Healthy	0.82 ± 0.09	0.80 ± 0.06	0.78 ± 0.14	0.59 ± 0.12	0.60 ± 0.17
	ALL/CLL	0.73 ± 0.10	0.62 ± 0.10	0.63 ± 0.12	0.46 ± 0.17	0.53 ± 0.12
Only Stimulation	Model type	Random forest	MLPC	Decision Tree	Nearest Neighbors	Naïve Bayes
	Healthy	0.29 ± 0.09	0.37 ± 0.12	0.29 ± 0.10	0.30 ± 0.10	0.41 ± 0.09

ALL/CLL	0.29 ± 0.10	0.34 ± 0.07	0.25 ± 0.12	0.23 ± 0.09	0.38 ± 0.10
---------	-------------	-------------	-------------	-------------	-------------

We have added this observation to the main text:

“However, the phenotype of the input T cells, and whether it came from a healthy donor or patient sample is also important – models trained only on stimulation dose were significantly worse in accuracy (Supplementary Table 4).” (lines 239-241).

This highlights that while stimulation dose is quite informative on its own, it is not wholly informative and is therefore unable to accurately predict the optimal CAR-T cell product by itself.

Data sharing: It is unclear why all of the data requires an MTA. Much of the data generated (e.g. flow cytometry values, % cytotoxicity assays, etc) are not identifiable and don't seem to even require controlled access. For example, feature values, CAR-T product labels and % cytotoxicity could easily be included in a supplementary excel sheet.

The authors should consider releasing their code and data analysis notebooks via Github or similar to support reproducibility of their work.

Thank you for catching this error. It was always our intention to share our code to ensure accessibility and encourage reproducibility of this work. The data and code are accessible on Github: [https://github.com/siddharthriyer/car t stimulation](https://github.com/siddharthriyer/car_t_stimulation). We have modified the data availability statement to reflect this:

“All computational analyses and data used for model training have been made available at: [https://github.com/siddharthriyer/car t stimulation](https://github.com/siddharthriyer/car_t_stimulation). Patient-related data that was collected but not shown in the paper might be subject to confidentiality. All other relevant data requests should be made to the corresponding author.” (lines 618-621).

Minor comments:

Methods line 535: What do the authors mean by a ‘T cell event’ here? Does that just refer to a T cell from a specific CAR-T product? Is the idea that each T cell is characterized by a vector of features measured by flow cytometry? What features were used? Then T cells ‘inherit’ the % cytotoxicity of the CAR-T product. They are clustered based on the flow features, and the clusters are ascribed that average % cytotoxicity of the constituent T cells? The authors never actually mention the value of k in the ms, though from figure 2e it appears to be ~80.

Thank you for the questions. Here, we defined T-cell event as a single cell from a specific CAR-T cell product as captured via flow cytometry. That’s exactly right - each event/T-cell is characterized by a series of T-cell features, obtained from conventional gating strategies. These features include: Day 8 T-cell fold expansion, Day 8 CD4:CD8 ratio, %EGFR+ of CD3+ T cells, %CD45RA+CCR7+ of CD4+ or CD8+ T cells, %CD45RA+CCR7- of CD4+ or CD8+ T cells, %CD45RA-CCR7+ of CD4+ or CD8+ T cells, %CD45RA-CCR7- of CD4+ or CD8+ T cells, %PD-1+Tim-3+ of CD4+ or CD8+ T cells, %PD-1+Tim-3- of CD4+ or

CD8+ T cells, %PD-1-Tim-3+ of CD4+ or CD8+ T cells, %PD-1-Tim-3- of CD4+ or CD8+ T cells, %CD25+ of CD4+ or CD8+ T cells, %CD137+ of CD4+ or CD8+ T cells.

That's correct. Each T-cell event is assigned an experimentally observed % cytotoxicity of the CAR-T cell product and then clustered based on the features. Each cluster is then given a % cytotoxicity, which was calculated based on the average of the % cytotoxicity of the constituent T cells. The optimal k was ~70 for both healthy donor and patient-derived samples.

To make the methods for the cytotoxicity and phenotype correlation section clearer, we have modified the text accordingly (lines 551-567).

Figure 3A – both axes are labeled PC1

Thank you for catching this mistake. We have corrected the PC axis label in Fig. 3a.

Figure 4 d and e, the features are not labeled

We intentionally did not label these features as our goal was to emphasize the importance of stimulation dose. We instead highlighted the five most important features following stimulation dose in Fig. 4f.

Line 81-82 is not a complete sentence.

Thank you for catching this error. We have modified the text to link the two sentences together:

“We produced anti-CD19 CAR-T cells using cells from all samples using a second-generation construct consisting of an extracellular anti-CD19 scFv linked to an intracellular 41BB signaling domain.” (lines 81-82).

Line 100: Tim-3 should be TIM-3

Thank you, we have corrected this error in the text (line 100).

Line 309: Missing . after GVHD

We have corrected the error in the main text.

Line 311: Did the authors mean portions rather than proportions?

Yes, we did indeed mean portions. Thank you for catching this error!

REVIEWERS' COMMENTS

Reviewer #1 (Remarks to the Author):

The authors have modified the manuscript in response to prior reviewer comments. In particular, the authors include additional in vivo experimental data from a second donor. I am well aware of the data from prior publications comparing CD28 to 41BB CAR-expressing T cells and this is exactly why it will be critical to confirm the validity of this approach using different CAR constructs. Nonetheless, I understand the view that this may be beyond to scope of the current manuscript.

Reviewer #2 (Remarks to the Author):

The reviewer thanks the authors for their answers, and the substantial improvement of their manuscript especially with the in vivo experiments. Please correct in sup Fig 5 legend (l. 85) "in animals" written twice, otherwise no more comments.

Reviewer #3 (Remarks to the Author):

The authors have thoroughly addressed my comments.

Point-by-Point Response to Reviewer's Comments (Zhang et al. formerly manuscript NCOMMS-22-16280A)

Please see below for our responses to each comment from every reviewer. The reviewer's comments are in regular typeface, and our replies in bold typeface.

REVIEWERS' COMMENTS

Reviewer #1 (Remarks to the Author):

The authors have modified the manuscript in response to prior reviewer comments. In particular, the authors include additional in vivo experimental data from a second donor. I am well aware of the data from prior publications comparing CD28 to 41BB CAR-expressing T cells and this is exactly why it will be critical to confirm the validity of this approach using different CAR constructs. Nonetheless, I understand the view that this may be beyond to scope of the current manuscript.

We thank this reviewer for their thorough feedback. We fully acknowledge the differences between CD28/CD3 ζ and 41BB/CD3 ζ signaling and agree that this should be investigated as a next step. We hope to pursue this, and the contribution of other intracellular signaling constructs to create more detailed yet generalizable models of T-cell stimulation and CAR-T cell function in the future.

Reviewer #2 (Remarks to the Author):

The reviewer thanks the authors for their answers, and the substantial improvement of their manuscript especially with the in vivo experiments. Please correct in sup Fig 5 legend (l. 85) "in animals" written twice, otherwise no more comments.

We thank this reviewer for their insights, particularly regarding benchmarking studies. The addition of this data has greatly strengthened our manuscript. Thank you for catching the error in Supplementary Fig. 5. We have removed the repeated text. Thank you.

Reviewer #3 (Remarks to the Author):

The authors have thoroughly addressed my comments.

We thank this reviewer for their feedback, particularly regarding the computational elements of the manuscript. The modifications have improved the clarity and hopefully long-term reproducibility of the methodologies employed in this work.